# GLOBAL COUNTERFACTUAL EXPLANATIONS ARE RELIABLE OR EFFICIENT, BUT NOT BOTH

## ABSTRACT

Counterfactual explanations have been widely studied in explainability, with a range of application dependent methods emerging in fairness, recourse and model understanding. The major shortcoming associated with these methods, however, is their inability to provide explanations beyond the local or instance-level. While many works touch upon the notion of a global explanation, typically suggesting to aggregate masses of local explanations in the hope of ascertaining global properties, few provide frameworks that are both reliable and computationally tractable. Meanwhile, practitioners are requesting more efficient and interactive explainability tools. We take this opportunity to investigate existing methods, improving the efficiency of Actionable Recourse Summaries (AReS), one of the only known global recourse frameworks, and proposing Global & Efficient Counterfactual Explanations (GLOBE-CE), a novel and flexible framework that tackles the scalability issues associated with current state-of-the-art, particularly on higher dimensional datasets and in the presence of continuous features. Furthermore, we provide a unique mathematical analysis of categorical feature translations, utilising it in our method. Experimental evaluation with real world datasets and user studies verify the speed, reliability and interpretability improvements of our framework.

## 1 INTRODUCTION

Counterfactual explanations (CEs) construct input perturbations that result in desired predictions from machine learning (ML) models (Verma et al., 2020; Karimi et al., 2020; Stepin et al., 2021). A key benefit of these explanations is their ability to offer recourse to affected individuals in certain settings (e.g., automated credit decisioning). Recent years have witnessed a surge of subsequent research, identifying desirable properties of CEs (Wachter et al., 2018; Barocas et al., 2020; Venkatasubramanian & Alfano, 2020), developing the methods to model those properties (Poyiadzi et al., 2020; Ustun et al., 2019; Mothilal et al., 2020; Pawelczyk et al., 2021), and understanding the weaknesses and vulnerabilities of the proposed methods (Dominguez-Olmedo et al., 2021; Slack et al., 2021; Upadhyay et al., 2021; Pawelczyk et al., 2022). Importantly, however, the research efforts thus far have largely centered around local analysis, generating explanations for individual inputs.

Such analyses can vet model behaviour at the instance-level, though it is seldom obvious that any of the resulting insights would generalise globally. For example, a local CE may suggest that a model is not biased against a protected attribute (e.g., race, gender), despite net biases existing. A potential way to gain such insights is to aggregate local explanations (Lundberg et al., 2020; Pedreschi et al., 2019; Gao et al., 2021), but since the generation of CEs is generally computationally expensive, it is not evident that such an approach would scale well or lead to *reliable* conclusions about a model's behaviour. Be it during training or post-hoc evaluation, global understanding ought to underpin the development of ML models *prior* to deployment, and reliability and efficiency play important roles therein. We seek to address this in the context of global counterfactual explanations (GCEs).

### 1.1 CONTRIBUTIONS: INVESTIGATIONS, IMPLEMENTATIONS & IMPROVEMENTS

Given the current lack of a precise definition, we posit in this work that a GCE should apply to multiple inputs simultaneously, while maximising accuracy across such inputs. For clarity, we distinguish *counterfactuals* (the altered inputs) from *counterfactual explanations* (the extension of counterfactuals to any of their possible representations, e.g., translation vectors, rules denoting fixed values, etc.).

**Investigations**  Section 2 summarises GCE research, introducing the recent Actionable Recourse Summaries (AReS) framework in Rawal & Lakkaraju (2020). We then discuss motivations, defining reliability and justifying our claim that current GCE methods are reliable or efficient, but not both.

Figure 1: Left: GLOBE-CE scaled translations. We argue that, while many translation directions cannot be interpreted, we can optimise GCEs by allowing variable magnitudes per input. Right: Example comparisons with synthetic *ForeignWorker* subgroups. Left to right: Accuracy-cost trade-offs (covering more inputs requires larger magnitudes), minimum costs per input, and the mean translation direction for each subgroup.

**Our Framework**  Section 3 proceeds to introduce our framework Global & Efficient Counterfactual Explanations (GLOBE-CE). Though not strictly bound to recourse, our framework has the ability, as in AReS, to seek answers to the big picture questions regarding a model's recourses (Figure 1), namely the potential disparities between affected subgroups, i.e. *Do race or gender biases exist in the recourses of a model? Can we reliably convey these in an interpretable manner?*

Our major contribution is a shift in paradigm; all research so far assumes GCEs to be fixed. We represent each GCE with a fixed translation vector $\underline{\delta}$, multiplied by an input-dependent, scalar variable $k$ (Figure 1). To determine the direction of each translation, our framework deploys a general CE generation scheme, flexible to various desiderata. These include but are not limited to sparsity, diversity, actionability and model-specific CEs (Section 2.1). However, the novelty of our method lies mainly in a) varying $k$ input-wise and b) proving that arbitrary translations on one-hot encodings can be expressed using If/Then rules. To the best of our knowledge, this is the first work that addresses mathematically the direct addition of translation vectors to one-hot encodings in the context of CEs.

**AReS Implementations**  Section 4 subsequently outlines our AReS and Fast AReS implementations, where in the latter we propose amendments to the algorithm and demonstrate that these lead to significant speed and performance improvements on four benchmarked financial datasets. Both implementations are thereafter utilised as baselines in the experimental evaluation of GLOBE-CE.

**Improvements**  Section 5 evaluates the efficacy of our Fast AReS and GLOBE-CE frameworks along three fundamental dimensions: *accuracy* (the percentage of inputs with successfully altered predictions), *average cost* (the difficulty associated with executing successful GCEs) and *speed* (the time spent computing GCEs). We argue that GCEs that fail to attain maximum accuracy or minimum cost can be misleading, raising concerns around the safety of ML models vetted by such explanations. We target these metrics, demonstrating significant speedups at concurrently higher accuracies and lower costs. User studies comprising ML practitioners additionally demonstrate the ability of the GLOBE-CE framework to more reliably detect recourse biases where previous methods fall short.

## 2  RELATED WORK: GLOBAL COUNTERFACTUAL EXPLANATIONS

### 2.1  LOCAL COUNTERFACTUAL EXPLANATIONS: INSTANCE-LEVEL MODEL INSIGHTS

Wachter et al. (2018) is one of the earliest introductions of CEs in the context of understanding black box ML models, defining CEs as points close to the query input (w.r.t. some distance metric) that result in a desired prediction. This inspired several follow-up works proposing desirable properties of CEs and presenting approaches to generate them. Mothilal et al. (2020) argues the importance of diversity, while other approaches aim to generate plausible CEs by considering proximity to the data manifold (Poyiadzi et al., 2020; Van Looveren & Klaise, 2021; Kanamori et al., 2020) or by accounting for causal relations among input features (Mahajan et al., 2019). Actionability of recourse is another important desideratum, suggesting certain features be excluded or limited (Ustun et al., 2019). In another direction, some works generate CEs for specific model categories, such as tree-based (Lucic et al., 2022; Tolomei et al., 2017; Parmentier & Vidal, 2021) or differentiable (Dhurandhar et al., 2018) models. Detailed surveys on CEs naturally follow (Karimi et al., 2020; Verma et al., 2020).

## 2.2 Beyond Local Counterfactual Explanations: The Curse of Globality

Despite a growing desire from practitioners for global explanation methods that provide summaries of model behaviour (Lakkaraju et al., 2022), the struggles associated with summarising complex, high-dimensional models globally is yet to be comprehensively solved. Some manner of local explanation aggregation has been suggested (Lundberg et al., 2020; Pedreschi et al., 2019; Gao et al., 2021), though no compelling results have been shown that are both reliable and computationally tractable for GCEs specifically. Lakkaraju et al. (2022) also indicates a desire for more interactivity with explanation tools, alongside reliable global summaries, but these desiderata cannot be paired until the efficiency issues associated with global methods are addressed in research.

Such works have been few and far between. Both Plumb et al. (2020) and Ley et al. (2022) have sought global translations which transform inputs within one group to another desired target group, though neither accommodate categorical features. Meanwhile, Becker et al. (2021) provides an original method, yet openly struggles with scalability. Gupta et al. (2019) attempts to equalize recourse across subgroups during model training, but with no explicit framework for global interpretation. Rawal & Lakkaraju (2020) proposes AReS, a comprehensive GCE framework which builds on previous work in (Lakkaraju et al., 2019). AReS adopts an interpretable structure, termed two level recourse sets. We defer details of our AReS implementation, improvements included, to Section 4. To our knowledge, only AReS and recent adaptation CET (Kanamori et al., 2022) pursue GCEs for recourse, yet the latter reports runtimes in excess of three hours for both methods. Since our goal is to provide practitioners with fast and reliable global summaries, potentially at each iteration of model training (Gupta et al., 2019), we must take steps to bridge the gap between reliability and efficiency.

## 2.3 Motivation: Global CEs Are Reliable Or Efficient, But Not Both

**Reliability** In this work, we define reliable GCEs to be those that can be used to draw accurate conclusions of a model's behaviour. For instance, a model with higher recourse costs for subgroup A than subgroup B is said to exhibit a *recourse bias* against subgroup A, and reliable GCEs would yield the *minimum* costs of recourse for either subgroup, such that this bias could be identified. If the costs found are sub-optimal, or GCEs only apply to a small number of points (sub-optimal accuracy), the wrong conclusions might be drawn, as in GCEs $A_i$, $B_i$ of Figure 2. In the absence of minimum cost recourses, biases may be detected where not present ($A_1$, $B_1$) or not detected where present ($A_2$, $B_2$). Similarly, without sufficient accuracy, the same phenomena may occur ($A_3$, $B_3$ and $A_4$, $B_4$, respectively). The further these metrics stray from optimal, the less likely any potential subgroup comparisons are of being reliable. We argue that maximising reliability thus amounts to maximising accuracy while minimising recourse costs. The wider scope of reliability may encompass other desiderata as outlined in Section 2.1, though we refer their impact on bias assessment to future work.

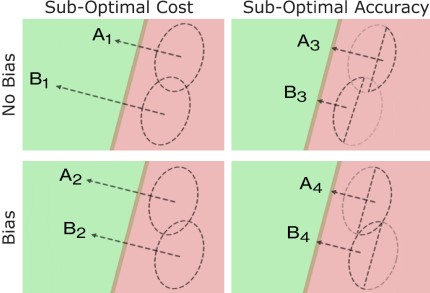

Figure 2: Common pitfalls with *unreliable* bias assessment ($\ell_2$ distance represents cost) for GCEs $A_i$, $B_i$. Light dotted lines: inputs from subgroups A and B. Dark dotted lines: inputs for which the respective GCEs apply.

**Efficiency** There exists a gap in GCE research between reliability and efficiency. Even the most comprehensive works that target maximum reliability (Rawal & Lakkaraju, 2020; Kanamori et al., 2022) suffer computation times in excess of three hours on relatively small datasets such as German Credit (Dua & Graff, 2019). In parallel, there exists a body of research advocating strongly the use of inherently interpretable models for these cases, where performance is not compromised (Rudin, 2019; Rudin & Radin, 2019; Chen et al., 2018). The utility in black box explanations is thus mainly reserved for higher complexity scenarios, and we must seek a global method that both executes efficiently and scales well, criteria by which current works have fallen short. In our experiments, we define efficiency in relation to the average CPU time taken in computing GCE explanations.

## 3 Our Framework: Global & Efficient CEs (GLOBE-CE)

We proceed to detail our proposed GLOBE-CE framework, where we discuss below: 1) the representation of GCEs that we choose, assisted by theoretical results on categorical feature arithmetic, 2) the GLOBE-CE algorithm, and 3) the adaptability of our framework to existing CE desiderata.

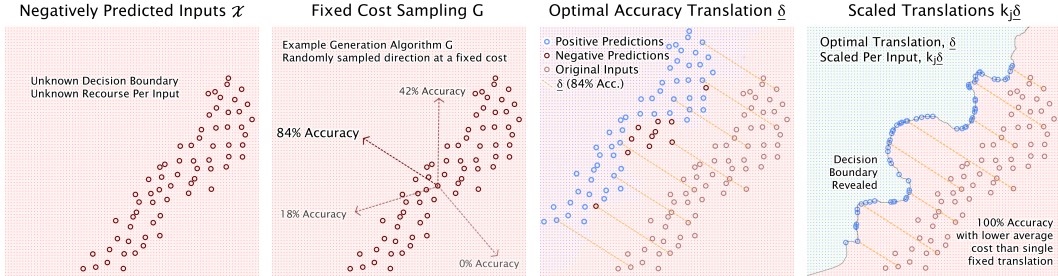

Figure 3: The GLOBE-CE framework (Algorithm 1) for **specific** $G$. Cost is $\ell_2$ distance. Left: Negative predictions, $\mathcal{X}$. Left Center: Fixed cost sampling, $G$. Right Center: Highest accuracy $\underline{\delta}$ selection. Right: Scaling $\underline{\delta}$ per input. Theorems 1 and 2 are essential in bridging the gap between scaling translations and the discontinuous nature of categorical features. Future work may consider scaling translations to a fixed end-point.

## 3.1 OUR GCE REPRESENTATION: SCALED TRANSLATION VECTORS

We propose a novel, interpretable GCE representation: scaled translation vectors, as depicted in Figures 1 and 3. Simply put, for inputs that belong to a particular subgroup $\underline{x} \in \mathcal{X}_{\text{sub}}$, we can apply a translation $\underline{\delta}$ with scalar $k$ such that $\underline{x}_{\text{CF}} = round(\underline{x} + k\underline{\delta})$ is a successful counterfactual. Note that $round(\underline{x})$ re-encodes one-hot outputs by selecting the largest feature values post-translation. For each $\underline{x} \in \mathcal{X}_{\text{sub}}$, our framework computes the respective minimum value of $k$ required for recourse. The main appeal of this approach is its improvement with respect to the interpretability-performance trade-off that other methods suffer from. Using too few translations limits the performance of previous methods, yet large numbers of GCEs cannot easily be interpreted. For instance, Figure 3 demonstrates conceptually how one can achieve maximum accuracy with a single translation at comparably lower average costs to previous methods which do not utilise variable magnitudes (Section 2.3 details why this is absolutely necessary for reliable bias assessment). While one can interpret a range of scalars relatively easily (Section 3.2), one cannot simply interpret a whole range of separate directions, and therein lies the paradigm shift that we propose.

We posit that our method of a) assigning a single vector direction to an entire subgroup of inputs, b) travelling along this vector, and c) analysing the minimum costs required for successful recourses per input of the subgroup, is the natural global extension to one of the simplest forms of local CE: the *fixed magnitude* translation. In fact, the connection between local and global explanations may be more intimate than current research implies. Works suggesting to learn global summaries from local explanations (Pedreschi et al., 2019; Lundberg et al., 2020; Gao et al., 2021) and approaches suggesting to learn global summaries directly (Rawal & Lakkaraju, 2020; Kanamori et al., 2022; Plumb et al., 2020; Ley et al., 2022) tend to approach global explanations from the angle that they are fundamentally different problems. Our framing of GCEs as a local problem is akin to treating groups of inputs as single instances, generating translations for them and subsequently, through scaling, efficiently capturing the true range of properties across the set of local instances and their proximity to the decision boundary (Figure 3).

Unlike prior work, this implies that we can tackle a wide range of minimum costs, and potentially complex, non-linear decision boundaries, despite a fixed direction $\underline{\delta}$. AReS (Rawal & Lakkaraju, 2020) and recent similar work Counterfactual Explanation Trees (CET) (Kanamori et al., 2022) do not propose any such scaling techniques. The latter indicates the accuracy to cost trade-off that occurs when one is limited by a *fixed* translation, yet eventually compromises for one. Other translation works (Plumb et al., 2020; Ley et al., 2022) do not utilise any form of scaling, nor do they provide steps to handle categorical features, an issue we address below. These approaches can also be prone to unreliability since they target training data rather than a model's decision boundary.

**Categorical Feature Arithmetic**   We assume one-hot encodings (else, these can be trivially encoded and decoded to suit) and provide, to the best of our knowledge, the first work in the context of CEs that reports mathematically the interpretation of a translation $k\underline{\delta}$ on a one-hot encoding, including the effects of scaling, yielding deterministic, interpretable rules from $k\underline{\delta}$. For examples of the interpretable rules sets generated by categorical translations, see Table 1 and Appendix B.2.

**Theorem 1.** *Regardless of feature value, any translation vector that is added to a one-hot categorical feature can alternatively be expressed using If/Then rules, with just one unique Then condition.*

*Proof (Sketch).* Consider any one-hot encoded feature vector with feature labels ranging from 1 to $n$, denoted $\underline{f} = [f_1, f_2, ..., f_n] \in \{0, 1\}^n$, where $|\underline{f}|_1 = 1$ and $F = \arg\max_i f_i$. Similarly, consider a translation vector of size $n$, denoted $\underline{\delta} = [\delta_1, \delta_2, ..., \delta_n] \in \mathcal{R}^n$, where $\Delta = \arg\max_i \delta_i$. The final vector post-translation is $\underline{g} = \underline{f} + \underline{\delta}$, and the final feature value is $G = \arg\max_i g_i$. Note $g_{i \neq F} = \delta_i$ and $g_F = \delta_F + 1$. We denote $g_G = \max_i g_i = \max(\delta_F + 1, \max_{i \neq F}(\delta_i))$. For $1 \leq F \leq n$, we now prove that if $G \neq F$ (i.e. a change in feature value occurs), we have the rule "If $F$, Then $\Delta$". In the case $F = \Delta$, $g_G = \max(\delta_\Delta + 1, \max_i(\delta_{i \neq \Delta})) = \delta_\Delta + 1$, as $\delta_\Delta = \max_i \delta_i$. Hence, $G = \Delta$ (no rule). In the case $F \neq \Delta$, $g_G = \max(\delta_F + 1, \delta_\Delta)$. If $\delta_F + 1 > \delta_\Delta$, then $g_G = \delta_F + 1$ and $G = F$ (no rule). However, if $\delta_F + 1 < \delta_\Delta$, then $g_G = \delta_\Delta$ and $G = \Delta$, giving the rule "If $F$, Then $\Delta$". □

**Theorem 2.** *Regardless of feature value, any translation vector that is scaled by $k \geq 0$ and added to a one-hot categorical feature can alternatively be expressed with the first $m$ rules of a sequence.*

*Proof (Sketch).* Consider the general vectors $\underline{f}$ and $\underline{\delta}$ defined in Theorem 1, and scalar $k$. For $i \neq \Delta$ and $k > 0$, Theorem 1 gives that $k\delta_i + 1 < k\delta_\Delta$ yields the rule "If $i$, Then $\Delta$". Rearranging gives that if the lower bound $k > \frac{1}{\delta_\Delta - \delta_i}$ is satisfied, then the translation $k\underline{\delta}$ induces such a rule. Consider additionally the vector of lower bounds $\underline{k} = [k_1, k_2, ..., k_n] \in \mathcal{R}_+^n$ where $k_{i \neq \Delta} = \frac{1}{\delta_\Delta - \delta_i}$ and $k_\Delta = \infty$.

**Lemma 2.1.** *By inspection, we have that $k_i \leq k_m$ for any $i, m < n$ pair with $\delta_i \leq \delta_m$. As such, lower bounds for $i$ and $m$ are both satisfied if $k > k_m$. Thus, scaling $\underline{\delta}$ by $k > k_m$ induces not only the rule corresponding to feature value $m$, but also that of any other feature value $i$ with $\delta_i \leq \delta_m$.*

For $k = 0$, we have no rules ($k\underline{\delta} = \underline{0}$). Let $\Delta_i$ now be the index of the $i^{\text{th}}$ smallest value in $\underline{\delta}$, such that $\Delta_1 = \arg\min_i \delta_i$ and $\Delta_n = \arg\max_i \delta_i = \Delta$. Thus, by Lemma 2.1, for $m < n$, we have that scaling $\underline{\delta}$ by $k_{\Delta_m} < k \leq k_{\Delta_{m+1}}$ induces rules for each of the first $m$ feature values $\Delta_{1 \leq i \leq m}$. □

## 3.2 THE GLOBE-CE ALGORITHM: LEARNING AND INTERPRETING TRANSLATIONS

Any particular CE may be represented with a *fixed magnitude* translation. The major contribution of the GLOBE-CE framework lies in the notion of *scaling the magnitudes* of translations. Though perhaps an uninteresting concept in the context of local CEs, when large numbers of inputs are present, scaling a translation $\underline{\delta}$ with an *input-dependent* variable $k$ is an elegant and efficient way of solving for global summaries of a model's decision boundary. Figure 3 depicts the utility that a single, scaled translation vector can exhibit. One can interpret a range of magnitudes, though cannot interpret a range of directions so easily, and previous approaches relied on using a small number of fixed GCEs.

**Learning Translations** In our setup, we learn explanations by adopting methods from instance-level CE research, generalising for any CE algorithm $G(B, \mathcal{X}, n)$ that considers, at a minimum, the model $B$ being explained, the inputs requiring explanations $\mathcal{X}$, and the number $n$ of returned GCEs $\underline{\delta}_1, \underline{\delta}_2, ..., \underline{\delta}_n = \Delta$. This enables previous CE research to simply be extended when seeking global explanations; we show in Section 3.3 how a previous method (constrained random sampling) can be adapted for our purposes. GLOBE-CE then scales the $i^{th}$ GCE $\underline{\delta}_i$

---

**Algorithm 1:** Flexible GLOBE-CE Framework

**Inputs:** $B$, $\mathcal{X}$, $G$, $n$, $\underline{k}$, $cost$

1: $\Delta = G(B, \mathcal{X}, n)$ ▷ Generate GCEs (Translations)
2: **for** $1 \leq i \leq n$ **do** ▷ For all GCEs
3:    **for** $1 \leq j \leq |\underline{k}|$ **do** ▷ For all Scalars
4:       $\mathcal{X}'_{ij} = round(\mathcal{X} + k_j \underline{\delta}_i)$ ▷ Counterfactuals
5:       $\mathcal{Y}'_{ij} = B(\mathcal{X}'_{ij})$ ▷ Predictions
6:       $\mathcal{C}_{ij} = cost(\mathcal{X}, \mathcal{X}'_{ij})$ ▷ Costs
7:    **end for**
8: **end for**

**Outputs:** Counterfactuals $\mathcal{X}'$, Predictions $\mathcal{Y}'$, Costs $\mathcal{C}$ (For all Inputs $\mathcal{X}$, Translations $\Delta$ and Scalars $\underline{k}$)

---

over a range of $m$ scalars $\underline{k} = k_1, k_2, ..., k_m$, repeating over all $1 \leq i \leq n$ GCEs and returning the counterfactuals $\mathcal{X}'$, the predictions $\mathcal{Y}' \in \{0, 1\}^{n \times m \times |\mathcal{X}|}$ and costs $\mathcal{C} \in \mathcal{R}_{\geq 0}^{n \times m \times |\mathcal{X}|}$.

In practice, one could set an upper limit of $n$ for the maximum number of GCEs end users are willing or able to interpret. Algorithm 1 could then be terminated when accuracies and costs plateau, or $n$ is reached. Scalars $\underline{k}$ are chosen in a similar fashion, with an upper limit on the cost of GCEs i.e. such that $\underline{k}$ simply ranges linearly from 0 to a point that yields this limit. This may vary per translation, a property not captured here, though one that is easily implementable. Given the speed of our method (Section 5), we found $m = 1000$ to be appropriate. For categorical features, scalars exhibit certain properties that can be manipulated (Theorem 2).

**Interpreting Translations** The manner in which explanations are portrayed depends on the nature of the data and/or the desire to compare recourses. We introduce straightforward interpretations of GCEs in continuous and categorical contexts, alongside the corresponding accuracy/cost properties.

| Feature(s) | New Rule Added | New Inputs | | All Inputs | |
|:---:|:---:|:---:|:---:|:---:|:---:|
| | | Accuracy | Cost | Accuracy | Cost |
| **Account Status** | If F2, Then F4 | +33.5% | 1.00 | 33.5% | 1.00 |
| **Account Status** | If F3, Then F4 | +2.5% | 1.00 | 36.0% | 1.00 |
| **Account Status** | If F1, Then F4 | +45.2% | 1.00 | 81.2% | 1.00 |
| **Telephone** | If F2, Then F1 | +2.5% | 1.80 | 83.7% | 1.02 |
| **Employment** | If Not F4, Then F4 | +10.2% | 1.95 | 93.9% | 1.12 |

Table 1: Example *Cumulative Rules Chart (CRC)* for categorical features, representing the optimal GLOBE-CE translation at 5 scalars. Rules are cumulatively added, resulting in an increase in accuracies and costs.

Our scaling approach induces *accuracy-cost profiles* as pictured in Figure 1, Left Centre, providing an interpretable method for the selection of a particular accuracy/cost combination, as well as for bias assessment. Intuitively, if the magnitude of a translation is 0 (when $k = 0$), its accuracy and cost are also 0; as the magnitude grows, more inputs successfully cross the decision boundary, resulting in an increase in accuracy and average cost. Accuracy/cost values can then be chosen or compared.

We also adopt standard statistical methods to convey *minimum costs*, which correspond to the minimum scalars required to alter each input's prediction. For continuous data, where costs scale linearly as a particular translation is scaled, we deem it interpretable to display solely *mean translations* alongside the illustration of minimum costs (Figure 1 and Appendix B.1), an assertion supported by our user studies, where participants interpreted GLOBE-CE explanations considerably faster than in AReS. Prior to our analysis, translations as raw vectors in input space lacked an immediate and intuitive interpretation on categorical data. Theorem 1 demonstrates that any translation can be interpreted as a series of If/Then rules, limited to one Then condition per feature, as portrayed by the individual rows in Table 1. Theorem 2 consequently proves that as a translation is scaled, If conditions are added to the rules for each feature (e.g., *Account Status* in Table 1). We name the resultant GCE representation a *Cumulative Rules Chart (CRC)*. See Appendix B.2 for further details.

### 3.3 ADAPTING THE GENERATION ALGORITHM FOR CE DESIDERATA & BIAS ANALYSIS

Recognising the scope of possible GCE generation algorithms to be vast, it should be stated that modifications to $G$ along arbitrary criteria may impact efficiency in ways not investigated herein.

**GCE Generation** In our context of recourse, $G(B, \mathcal{X}, n)$ should deliver *diverse* and relatively *sparse* translations, targeting reliability (maximum accuracy, minimum cost) and taking into account that increasing diversity will likely reduce interpretability. Regarding actionability, we assign costs feature-wise (Section 5). We propose a specific generation algorithm $G(B, \mathcal{X}, n, n_s, c, n_f, p)$ for our experiments, which consists of uniform random sampling of $n_s$ translations at a *fixed* cost $c$. The additional parameters control the number of randomly chosen features $n_f$, and the power $p$ to which random samples between 0 and 1 are raised, offering control over sparsity. The $n$ final GCEs are chosen to greedily maximise accuracy. There also exist model categories that provide alternative candidate $G$: model gradients in Deep Neural Networks (DNNs), feature attributions in XGBoost (XGB) models, and the mathematics of Support Vector Machines (SVMs) are explored in Appendix B.3.

**Bias Analysis** Extending our framework to provide comparisons between affected subgroups of interest, similarly to AReS, is a trivial matter of separating the inputs and evaluating and scaling translations separately. However, it is recommended to generate the same set of $n_s$ random translations for both subgroups as this a) eliminates any possible random bias in our translation generation and b) executes faster. Alternatively, the translations selected for each subgroup can be exchanged, such that recourses can be directly compared (functionality that previous work does not suggest).

### 4 IMPROVING ACTIONABLE RECOURSE SUMMARIES: FAST AReS

We preface the evaluation of GLOBE-CE with an introduction to AReS (Rawal & Lakkaraju, 2020), before suggesting improvements to the method such that both AReS and our improved version can be used as baselines. AReS adopts an interpretable structure, termed two level recourse sets, comprising of triples of the form Outer-If/Inner-If/Then conditions (depicted in Figure 4). Subgroup descriptors $\mathcal{SD}$ (Outer-If conditions) and recourse rules $\mathcal{RL}$ (frequent itemsets output by apriori (Agrawal & Srikant, 1994)) determine such triples. Iteration over $\mathcal{SD} \times \mathcal{RL}^2$ yields the ground set $V$, before

| (Stage 1) Ground Set Generation | | (Stage 2) Ground Set Evaluation | (Stage 3) Ground Set Optimisation |
|---|---|---|---|
| *Frequent Itemsets* $\mathcal{SD}, \mathcal{RL}$ ⋮ | *Ground Set* $V \subset \mathcal{SD} \times \mathcal{RL}^2$ *(Valid Triples)* ⋮ | *Evaluate all triples in* $V \subset \mathcal{SD} \times \mathcal{RL}^2$ | *While there exists a delete, add or exchange operation that increases the value of the solution set* $f(\mathcal{R})$, *and satisfies constraints, perform it:* |

Foreign-Worker = True
Foreign-Worker = False
0 <= Years-At-Current-Home < 1
1 <= Years-At-Current-Home < 2
Sex = Male, 20 <= Age < 30
Sex = Male, 30 <= Age < 40

*If* Foreign-Worker = True:
 *If* Sex = Male and 20 <= Age < 30,
 *Then* Sex = Male and 30 <= Age < 40, $\Big\}\ v_i$

*If* Sex = Male and 20 <= Age < 30:
 *If* 0 <= Years-At-Current-Home < 1,
 *Then* 1 <= Years-At-Current-Home < 2 $\Big\}\ v_{i+1}$

*Initialise the solution* $\mathcal{R}$ *as the singleton set with the largest objective function value:*

$$v_{\max} = \arg\max_{v \in V} f(\{v\})$$

$$\mathcal{R} = \{v_{\max}\}$$

*Delete: remove a triple from* $\mathcal{R}$
*Add: move a triple from* $V$ *to* $\mathcal{R}$
*Exchange: swap a triple in* $\mathcal{R}$ *with one from* $V$

Figure 4: Workflow for our AReS implementation (bar improvements). Iteration over $\mathcal{SD} \times \mathcal{RL}^2$ computes all valid triples (Outer-If/Inner-If/Then conditions) in the ground set $V$ (Stage 1). $V$ is evaluated itemwise (Stage 2), and the optimisation Lee et al. (2009) is applied (Stage 3), returning the smaller two level recourse set, $R$.

submodular maximization algorithm (Lee et al., 2009) computes the final set $R \subseteq V$ using objective $f(R)$. One strength of AReS is in assessing fairness via disparate impact of recourses, through user-defined $\mathcal{SD}$. While a novel method with an interpretable structure, AReS can fall short on two fronts:

**Computational Efficiency** Our analyses suggest that AReS is highly dependent on the cardinality of the ground set $V$, resulting in impractically large $V$ to optimise. Our amendments efficiently generate denser, higher-performing $V$, unlocking the utility practitioners have expressed desire for.

**Continuous Features** Binning continuous data prior to GCE generation, as in AReS, struggles to trade speed with performance: too few bins results in unrealistic recourses; too many bins results in excessive computation. Our amendments demonstrate significant improvements on continuous data.

The overall search for a two level recourse set $R$ can be partitioned into three stages, as detailed in Figure 4 and Table 10. We generate $V$, evaluate $V$, and optimise $V$, to return a smaller, interpretable set $R \subset \mathcal{SD} \times \mathcal{RL}^2$. As in the original work, we set $\mathcal{SD} = \mathcal{RL}$ and let $|\mathcal{RL}| = n \Rightarrow |V| < n^3$. Our optimisations include: $\mathcal{RL}$-*Reduction* and *Then-Generation*, which V faster or generate higher performing V, respectively (Stage 1); *V-Reduction*, a method which terminates evaluation of $V$ after $r$ or $r'$ iterations, (Stage 2); and *V-Selection* which selects the best $s$ items from the ground set (Stage 3). Please refer to Appendix C for complete details and justifications.

## 5 EXPERIMENTS

The efficacy of our proposed Fast AReS and GLOBE-CE frameworks are evaluated herein. We analyse in Section 5.1 the Fast AReS performance enhancements, before providing in Sections 5.2 and 5.3 an extensive evaluation of our GLOBE-CE framework against the current state-of-the-art, AReS (and Fast AReS), and a user study involving 16 ML practitioners tasked with assessing recourse bias in models using either AReS or GLOBE-CE. The study also uncovers some important criteria for bias assessment. For further experimental specifics, please refer to Appendices A and D.

**Datasets** We employ four real world datasets to assess our methods, detailed in Table 2. Top down, these predict recidivism, credit risk, payment defaults, and credit risk. AReS (Rawal & Lakkaraju, 2020) utilised COMPAS (Larson et al., 2016) and German Credit (Dua & Graff, 2019); we include these in order to verify similar results, and we also introduce the larger Default Credit (Dua & Graff, 2019) and HELOC (FICO, 2018) datasets, which both include a significant number of continuous features.

| Dataset | $N$ | $D$ | $n_{cat}$ | $n_{cont}$ |
|---|---|---|---|---|
| COMPAS | 6k | 15 | 4 | 2 |
| German | 1k | 71 | 17 | 3 |
| Default | 30k | 91 | 9 | 14 |
| HELOC | 10k | 23 | 0 | 23 |

Table 2: Inputs $N$, Dimensions $D$, Categorical $n_{cat}$, Continuous $n_{cont}$.

**Models** We train 3 types of model: DNN, XGB, and Logistic Regression (LR). Model parameters are not held constant, rather the best combination for each dataset and model is chosen. We elect to train models on 80% of the data, unlike Rawal & Lakkaraju (2020). Given the class label imbalance in German Credit (30:70) and the size of the dataset ($N = 1000$), training AReS on 50% of the data is likely to yield only $0.5 \times 1000 \times 0.3 = 150$ negative predictions on which to construct recourses.

**Set-Up** AReS suggests binning continuous features, and specifying the cost of moving between two adjacent bins to be 1. We bin continuous features into 10 equal intervals post-training (see Section 4 trade-off). We also take the cost of moving from one categorical feature to another to be 1. As in AReS, we take the interpretability constraints $\epsilon_1, \epsilon_2, \epsilon_3 = 20, 7, 10$, and set $\mathcal{SD} = \mathcal{RL}$. Importantly, specifying subgroups simplifies the problem significantly, and so in our attempt to stress test

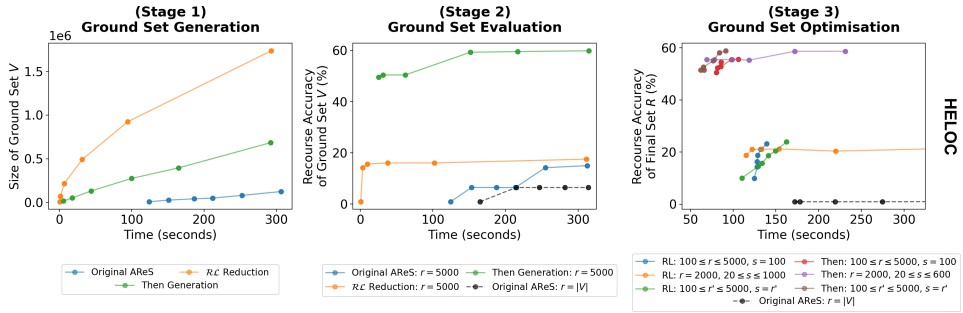

Figure 5: Fast AReS improvements (HELOC). Left: Size of ground set $V$ vs time. Centre: Accuracy of $V$ vs time. Right: Final set $acc(R)$ vs time. For other *categorical* datasets, $\mathcal{RL}$-*Reduction* achieved speedups similar to *Then-Generation*. For Default Credit (predominantly continuous), the latter performed best.

these explanation methods and fully gauge their scalability, we will avoid this. We also observed that both methods elected to alter a smaller number of features (2 or 3), regardless of any width limits.

## 5.1 FAST AReS EXPERIMENTAL IMPROVEMENTS

The evaluation of Fast AReS is broken down per workflow stage. For various hyperparameter combinations ($r$, $r'$ and $s$), the final sets returned in Stage 3 achieve significantly higher performance within a time frame of 300 seconds, achieving accuracies for which AReS required over 18 hours on HELOC. Appendix D lists details in full, including the combinations of hyperparameters used.

**Reliability & Efficiency** In Stage 1, we demonstrate $\mathcal{RL}$-*Reduction* is capable of generating identical ground sets orders of magnitude faster, and *Then Generation* constructs (different) ground sets rapidly. Stage 2 shrinking ($r = 5000$) significantly outperforms full evaluation, and *Then Generation* erases many continuous data limitations by short-cutting the generation of relevant rules. Finally, Stage 3 demonstrates vast speedups, owing to the generation of very small yet high-performing ground sets in the previous stage: $r$, $r'$ and $s$ restrict the size of $V$ yet retain a near-optimal $acc(V)$.

## 5.2 GLOBE-CE EXPERIMENTAL IMPROVEMENTS

The GLOBE-CE explanations in our experiments are computed by the generation procedure posed in Section 3.3, and their interpretations are as detailed in Section 3.2 (and Appendix B). We test the performance of both a single, scaled $\delta$, as well as the diverse solution outlined previously, with $n = 3$, denoting these GLOBE-CE and dGLOBE-CE, respectively. Translations are uniformly sampled at cost 2 and scaled in the range $0 \leq k \leq 5$, such that the maximum possible cost of a translation is 10 bins (the entire range for any one given feature), which we assume as a feasibility limit for recourse.

**Reliability** Evaluation over various families of models and four diverse datasets indicates that, across the board, GLOBE-CE consistently exhibits the most reliable performance. Situations in which only moderate accuracy is achieved by GLOBE-CE (e.g., COMPAS, DNN), we attribute to an inability of the model to provide recourses within our feasibility limit (after having exhaustively trialled solutions). In one rare case where the Fast AReS optimisation achieves superior cost (COMPAS, DNN), it does so at the expense of a significantly larger drop in accuracy (a natural trade-off).

**Efficiency** Applying AReS for several hours can yield accuracy improvements, though we consider such time scales inappropriate given that a) we test relatively simple datasets (Section 2.3) and b) practitioners are requesting higher levels of interactivity in explainability tools (Lakkaraju et al., 2022). We argue too that the concept of sampling random translations at a fixed cost is more intuitive than tuning hyperparameters of terms associated with cost, and that GLOBE-CE thus grants a higher degree of interactivity, given its performance with respect to computation time.

## 5.3 GLOBE-CE USER STUDIES

We conduct an online user study to analyse and compare the efficacy of GLOBE-CE and AReS in detecting recourse biases. The user study involves 16 participants, all of which have a background in ML and some knowledge of post-hoc explainability. We include a short tutorial on CEs, GCEs, and the ideas of recourse cost and recourse bias. The study utilises two black box models: the first is a decision tree with a *model* bias against females, though with a *recourse* bias exhibited against males

| Algorithms | Datasets | | | | | | | | | | | |
|---|---|---|---|---|---|---|---|---|---|---|---|---|
| | COMPAS | | | German Credit | | | Default Credit | | | HELOC | | |
| | Acc. | Cost | Time | Acc. | Cost | Time | Acc. | Cost | Time | Acc. | Cost | Time |
| **DNN** AReS | 51% | 2.31 | 101s | 73% | 1.6 | 2712s | 7.22% | 1.0 | 7984s | 5.4% | 1.0 | 9999s |
| Fast AReS | 64% | **1.45** | 32.0s | 72% | 1.43 | 12.8s | 99.8% | 4.2 | 37.3s | 52% | 5.5 | 109.1s |
| GLOBE-CE | 66% | 1.53 | **7.08s** | 85% | 1.2 | **2.28s** | 98.5% | 1.3 | **3.6s** | 93% | 4.3 | **4.66s** |
| dGLOBE-CE | **70%** | 1.46 | 9.15s | **90%** | **1.1** | 2.63s | **100%** | **1.1** | 7.86s | **95%** | 3.8 | 5.46s |
| **XGB** AReS | 45% | 1.9 | 205s | 61% | 1.5 | 2092s | 11% | 1.0 | 9999s | 1.7% | 1.0 | 9999s |
| Fast AReS | 83% | 1.9 | 47.6s | 65% | 1.75 | 34.33s | 93% | 2.3 | 29.97s | 28% | **2.1** | 93.58s |
| GLOBE-CE | 78% | 1.8 | **9.61s** | **95%** | **1.02** | **5.04s** | 96% | 1.1 | **2.94s** | 58% | 2.4 | **4.7s** |
| dGLOBE-CE | **91%** | **1.4** | 12.4s | 83% | 1.03 | 5.95s | **100%** | **0.7** | 6.35s | **80%** | 2.4 | 5.6s |
| **LR** AReS | 79% | 1.5 | 506s | 85% | 1.3 | 3566s | 31% | 1.2 | 9999s | 4.8% | 1.0 | 9999s |
| Fast AReS | 82% | 1.7 | 43.0s | 85% | 1.3 | 9.3s | 99% | 2.1 | 17.82s | 92% | 1.6 | 127.3s |
| GLOBE-CE | 83% | 1.20 | **8.43s** | 82% | **1.2** | **3.39s** | **100%** | **1.0** | **3.42s** | **100%** | **0.5** | **3.11s** |
| dGLOBE-CE | **84%** | **1.18** | 11.7s | **91%** | 1.3 | 3.87s | **100%** | **1.0** | 7.21s | **100%** | **0.5** | 3.85s |

Table 3: Evaluating the reliability (accuracy/cost as per Section 1.1) and efficiency of GLOBE-CE against AReS (Rawal & Lakkaraju, 2020). We have highlighted in red explanations that a) achieve below 10% accuracy or b) require computation time in excess of 10,000 seconds (≈3 hours). Best metrics are shown in **bold**.

due to the nature of the data distribution; the second is an SVM (where theoretical minimum $\ell_2$ costs can be computed as a ground truth) with a recourse bias against a synthetic *ForeignWorker* subgroup.

We randomly group the 16 participants into two equal subgroups, whereby each participant is presented with two global explanations generated from either AReS or GLOBE-CE. For each explanation, the user study asks two questions: 1) **do you think there exists bias in the presented recourse rules?**, and 2) **explain the reasoning behind your choice.** The first question is multiple choice with three answers, and the second is descriptive. Table 4 details the response breakdown. Importantly, 7 of the 8 AReS users also found describing the recourse bias to be hard. Details of the models, recourses, and snapshots of the user study can be found in Appendix D.3.

| User Studies | AReS | | GLOBE-CE | |
|---|---|---|---|---|
| **Breakdown** | Bias? | Correct? | Bias? | Correct? |
| Black Box 1 | 7 | 0 | 7 | 7 |
| Black Box 2 | 1 | 0 | 5 | 4 |

Table 4: Bias detection results from user studies. Bias and correct columns: no. of users that identified a bias and no. of users that described it correctly, respectively.

**Takeaway** In summary, the study demonstrates: 1) correct bias identification requires an understanding of the underlying distribution of recourses (the proportion of inputs each rule applies to), 2) sub-optimal cost yields misleading conclusions and 3) GLOBE-CE both provides distribution information (AReS uncovers the biased *model*, yet misleads users with regards to the biased *recourses*), and outperforms AReS on cost with minimum recourse costs close to the ground truth SVM.

## 6 CONCLUSION

This work studies the current state of GCE methods, and addresses in detail the issues associated within the recently proposed AReS framework in Rawal & Lakkaraju (2020). We investigate works on both global and local counterfactual explanations before implementing and improving AReS. With mounting desire from a practitioner viewpoint for access to fast, interactive explainability tools (Lakkaraju et al., 2022), it is crucial that such methods are not inefficient. We propose improvements to the AReS framework that speed up the generation of GCEs by orders of magnitude, also witnessing significant accuracy improvements on continuous data. We further propose a novel GCE framework, GLOBE-CE, that further improves on the issues faced by AReS. Extensive experiments with four public datasets and user studies demonstrate the efficacy of our proposed framework in generating accurate global explanations that assist in identifying recourse biases. In future we hope to apply the proposed GLOBE-CE approach to other real world use cases, and conceive of further generation algorithms that improve its performance beyond that demonstrated here. Appendix E provides an at length discussion of such work, and the relation of our method to current issues facing local CEs such as robustness (Dominguez-Olmedo et al., 2021) and sensitivity (Slack et al., 2021), as well as a detailed discussion of the limitations of our work. We hope that the work here inspires further research into the particularly under-studied area of GCEs, and proves useful as the development of explainability tools grows in the coming years.

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

APPENDIX

This appendix is formatted as follows.

1. We discuss the *Datasets & Models* used in our work in Appendix A.
2. We discuss *Implementation Details* and *Example Outputs* of GLOBE-CE in Appendix B.
3. We discuss the *Implementation Details* for AReS and Fast AReS in Appendix C.
4. We list the *Experimental Details* of our work and analyse *Further Results* in Appendix D.
5. We discuss *Limitations* and areas of *Future Work* for GLOBE-CE in Appendix E.

Where necessary, we provide discussion for potential limitations of our work and future improvements or avenues for exploration.

## A   DATASETS & MODELS

Four benchmarked datasets are employed in our experiments, all of which describe binary classification and are publicly available. Details are provided below and in Table 5. Our experiments utilise three types of models: Deep Neural Networks (DNNs); XGBoost (XGB); and Logistic Regression (LR), described below and in Tables 6 through 8. Our user study also investigates linear kernel Support Vector Machines (SVMs), as these provide mathematical forms for minimum recourse.

### A.1   DATASETS

To assess our methods, we utilise the real world datasets detailed below and in Table 2. Where necessary, we augment input dimensions with one-hot encodings over necessary variables (e.g. Sex).

The **COMPAS** (Correctional Offender Management Profiling for Alternative Sanctions) dataset (Larson et al., 2016) classifies **recidivism risk**, based off of various factors including race, and can be obtained from and is described at: `https://www.propublica.org/article/how-we-analyzed-the-compas-recidivism-algorithm`. This dataset tests performance of our method in low-dimensional and highly-categorical settings.

The **German Credit** dataset (Dua & Graff, 2019) classifies **credit risk** and can be obtained from and is described at: `https://archive.ics.uci.edu/ml/datasets/statlog+(german+credit+data)`. The documentation for this dataset also details a cost matrix, where false positive predictions induce a higher cost than false negative predictions, but we ignore this in model training. Note that this dataset, which tests mainly categorical settings, is distinct from the common Default Credit dataset, described hereafter.

The **Default Credit** dataset (Dua & Graff, 2019) classifies **default risk** on customer payments and is obtained from and described at: `https://archive.ics.uci.edu/ml/datasets/default+of+credit+card+clients`. This dataset stress-tests scalability (increased inputs, dimensions and continuous features).

The **HELOC** (Home Equity Line of Credit) dataset (FICO, 2018) classifies **credit risk** and can be obtained from (upon request) and is described in detail at: `https://community.fico.com/s/`

| Name | No. Inputs | Input Dim. | Categorical | Continuous | No. Train | No. Test |
|---|---|---|---|---|---|---|
| COMPAS | 6172 | 15* | 4 | 2 | 4937 | 1235 |
| German Credit | 1000 | 71* | 17 | 3 | 800 | 200 |
| Default Credit | 30000 | 91* | 9 | 14 | 24000 | 6000 |
| HELOC | 9871 | 23 | 0 | 23 | 7896* | 1975* |

*Denotes values post-processing (one-hot encoding inputs, dropping inputs).

Table 5: Summary of the datasets used in our experiments. Although German Credit includes continuous features, we find that they have limited effect on the model both during training and in the resulting explanations.

`explainable-machine-learning-challenge`. We drop duplicate inputs, and inputs where all feature values are missing (negative values), and replace remaining missing values in the dataset with median values. Notably, the majority of features are monotonically increasing/decreasing.

## A.2 MODELS

We train models with an 80:20 split for each dataset. While each model's parameters differ, the universal aims are to a) achieve sufficient accuracy, b) avoid overfitting, and c) predict a similar class balance to the original data. The true proportion of negative labels in the training data of each dataset are **45%**, **30%**, **22%**, and **53%**, respectively. The Table 5 No. Train column dictates $|\mathcal{X}|$ in Tables 6 through 8, and the inputs in $|\mathcal{X}|$ with negative predictions from a particular model are denoted $|\mathcal{X}_{\text{aff}}|$.

**Deep Neural Networks (DNNs)**     We use the common library `PyTorch` to construct and train DNNs. Widths and depths of such models are outlined in Table 6. Layers include dropout, bias and ReLU activation functions. We map the final layer to the output using softmax, and use Adam to optimise a cross-entropy loss function. Table 6 details various model parameters/behaviours.

| Name | Width | Depth | Dropout | Train Acc. | Test Acc | $|\mathcal{X}_{\text{aff}}|$ | $|\mathcal{X}_{\text{aff}}|/|\mathcal{X}|$ |
|---|---|---|---|---|---|---|---|
| COMPAS | 30 | 5 | 0.4 | 68% | 65% | 2552 | **52%** |
| German Credit | 50 | 10 | 0.3 | 84% | 78% | 243 | **30%** |
| Default Credit | 80 | 5 | 0.3 | 81% | 81% | 5232 | **22%** |
| HELOC | 50 | 5 | 0.5 | 74% | 74% | 4334 | **55%** |

Table 6: Summary of the DNNs used in our experiments.

**XGBoost (XGB)**     Implementation from the common `xgboost` library.

| Name | Depth | Estimators | $\gamma, \alpha, \lambda$ | Train Acc. | Test Acc | $|\mathcal{X}_{\text{aff}}|$ | $|\mathcal{X}_{\text{aff}}|/|\mathcal{X}|$ |
|---|---|---|---|---|---|---|---|
| COMPAS | 4 | 100 | 1, 0, 1 | 70% | 68% | 2008 | **41%** |
| German Credit | 6 | 500 | 0, 0, 1 | 95% | 74% | 197 | **25%** |
| Default Credit | 10 | 200 | 2, 4, 1 | 90% | 83% | 3744 | **16%** |
| HELOC | 6 | 100 | 4, 4, 1 | 77% | 74% | 4323 | **55%** |

Table 7: Summary of the XGB models used in our experiments.

**Logistic Regression (LR)**     Implementation from the common `sklearn` library.

| Name | Max Iterations | Class Weights (0:1) | Train Acc. | Test Acc | $|\mathcal{X}_{\text{aff}}|$ | $|\mathcal{X}_{\text{aff}}|/|\mathcal{X}|$ |
|---|---|---|---|---|---|---|
| COMPAS | 1000 | 1:1 | 67% | 65% | 1940 | **39%** |
| German Credit | 1000 | 1:1 | 79% | 76% | 180 | **23%** |
| Default Credit | 2000 | 0.65:0.35 | 81% | 83% | 3858 | **16%** |
| HELOC | 2000 | 1:1 | 73% | 75% | 4282 | **54%** |

Table 8: Summary of the LR models used in our experiments. Class Weights refer to the loss function used.

## B   OUR FRAMEWORK: GLOBAL & EFFICIENT CES

This Appendix discusses several specificities concerning our methodology. Similarly to our AReS implementations, we acknowledge that there does exist scope to improve upon the efficiency of our method, though are encouraged by the superior results GLOBE-CE achieves relative to baselines.

## B.1 EXAMPLE GLOBE-CE REPRESENTATIONS FOR CONTINUOUS DATA (HELOC)

This section expands upon the various outputs of our framework (Section 3.2) in the context of continuous data from the HELOC dataset (FICO, 2018). We consider the following useful representations of the information computed in Algorithm 1: accuracy-cost profiles, minimum costs, and mean translations.

**Accuracy-Cost Profiles**   The natural trade-off associated with CEs is that between accuracy and cost; lower cost CEs are likely to cover fewer inputs, and vice versa. The literature typically accounts for this by introducing a hyperparameter (e.g. $\lambda$) to tune the trade-off. This however assumes a linear relationship between the metrics i.e. $\lambda$ is held constant during optimisation. Not only is this unrealistic, as the relative importance of accuracy normally varies with cost, but it introduces tuning difficulties for practitioners, i.e., it must be determined either *a priori* or through hyperparameter search which accuracy-cost combination is optimal, and subsequently which $\lambda$ value maps to this particular combination (non-trivial). As stated in Section 3.2, our scaling approach instead induces *accuracy-cost profiles* as in Figure 1, providing a far more interpretable method for accuracy/cost selection.

**Minimum Costs & Mean Translations**   Standard statistical methods can be adopted to convey the *minimum costs* per input (corresponding to the minimum scalar required to alter the prediction). For continuous data, knowing that costs scale linearly as a particular translation is scaled, we deem it interpretable to display solely *mean translations* alongside the illustration of minimum costs, an assertion supported by our user studies, where participants interpreted GLOBE-CE explanations considerably faster than in AReS. These explanation representations are depicted in Figure 1.

## B.2 EXAMPLE GLOBE-CE REPRESENTATIONS FOR CATEGORICAL DATA (GERMAN)

We now proceed to examine typical GLOBE-CE outputs for categorical features, using the German Credit (Dua & Graff, 2019) dataset. Of particular interest is the representation of scaled categorical translations. The accuracy-cost profile and minimum cost histogram naturally follow.

**Cumulative Rules Charts for Scaled Translations**   For the sake of our analysis, consider purely the categorical features from German Credit (see Section 3.2 for the introduction to CRCs).

Recall that, under a given translation direction, there exists a minimum cost, or equivalently a minimum scalar, at which recourse is achieved (shown in blue in Figure 6), if indeed recourse can be achieved at all. However, unlike continuous features, representing the range of potential minimum scalars across all inputs in an interpretable manner is not trivial. To generate the CRC in Table 1, we exploit the vector of lower bounds $\underline{k} = [k_1, k_2, ..., k_n]$ introduced by Theorem 2, which defines the minimum scalars required to generate specific feature value rules for a given translation. We select 5 equally spaced scalars from this vector (shown in Table 1 and in orange in Figure 6). The constrained selection of $n$ scalars from $\underline{k}$, such that minimum costs are achieved over the inputs where recourse is found in Algorithm 1, can be viewed as a *monotonic submodular maximisation*. Problems of this type are NP-hard, though unlike AReS, our required search spaces $\underline{k}$ have shown empirically to be of significantly reduced size. Lastly, the accuracy-cost and minimum costs properties of AReS are also shown in green for reference.

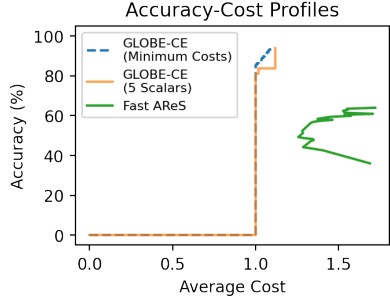

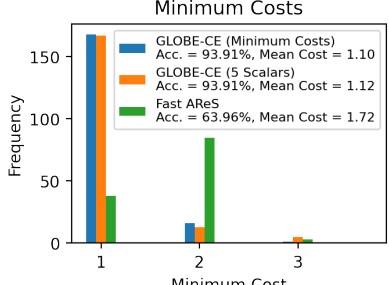

Figure 6: GLOBE-CE representations.

**Accuracy-Cost Profiles and Minimum Costs**   The numerical values in such a representation are easily portrayed via the *Accuracy-Cost Profiles* and *Minimum Costs Histograms* (Figure 6, Upper and Lower, respectively). The figures shown demonstrate our framework's superiority over AReS (including efficiency being around 400 times higher in this situation), even in the context of categorical features, which AReS is designed for, favouring the use of the categorical translation theorems. As seen, these representations are particularly useful in comparing recourses between subgroups– optimal translations for one subgroup can also be directly applied to another, and vice versa, to gain insights as to a model's subgroup-level reasoning. Potentially, the

rows of a CRC could too be reordered to achieve optimality. We discuss these future avenues for research in Appendix E.1.

**Notes Regarding Scaled Translations**   By our definitions in Section 5 and Appendix E, the cost of a scaled translation $k\underline{\delta}$ on an input $\underline{x}_1$ increases monotonically with $k$, as displayed in Figure 7 (see Theorem 2 for proof w.r.t. categorical features). However, between inputs $\underline{x}_1$ and $\underline{x}_2$, a smaller scaled translation $k_1\underline{\delta}$ may induce a higher cost on input $\underline{x}_1$ than a larger scaled translation $k_2\underline{\delta}$ induces on input $\underline{x}_2$ (Figure 7). For a fixed $\underline{\delta}$, inputs $\underline{x}_1$ and $\underline{x}_2$ can have different costs. For example, should an 'If' condition be satisfied by $\underline{x}_1$ that is not satisfied by $\underline{x}_2$, and $\underline{x}_1$ and $\underline{x}_2$ are otherwise the same, then $\underline{x}_1$ will have a greater cost than $\underline{x}_2$ (Figure 7). These last statements apply if and only if categorical features are involved.

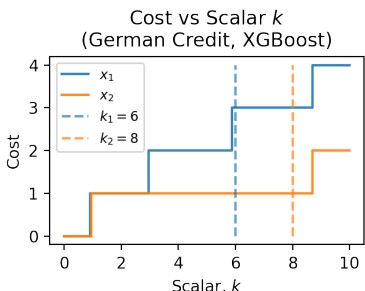

Figure 7: Display of a) monotonicity, b) smaller $k$ inducing higher cost, and c) fixed $k$ inducing variable cost.

**Scaled Translation Example (Single Categorical Feature)**   We conclude this section by providing one further example to aid understanding of the scaling process. Take a one-hot encoded feature with $n = 4$ values, translation $\underline{\delta} = [\delta_1, \delta_2, \delta_3, \delta_4] = [0, -1, 1, 1/2]$, index sequence of sorted $\delta$ (ascending) $\underline{\Delta} = [\Delta_1, \Delta_2, \Delta_3, \Delta_4] = [2, 1, 4, 3]$, vector of lower bound scalars $\underline{k} = [k_1, k_2, k_3, k_4] = [1, 1/2, \infty, 2]$. Recall that all rules will have the form 'If Feature Value = X, Then Feature Value = 3', since $\underline{\delta}_3$ is the maximum translation (Theorem 1). For example, $\underline{k}$ states intuitively that $\underline{\delta}$ must be scaled by a minimum factor of $k = 2$ in order for the translation to result in a rule for the fourth feature value, since $k_4 = 2$. If $k < 2$, inputs belonging to the fourth feature value will not flip value post-translation (take $k = 1$, where the post-translation value for such inputs is 1 for the third input and 1.5 for the fourth input, resulting in no change after re-encoding values by selecting the maximum). Table 9 illustrates the rule generation process in this case.

|  | $m = 1$ | $m = 2$ | $m = 3$ |
|---|---|---|---|
| $\Delta_m$ | 2 | 1 | 4 |
| $k_{\Delta_m}$ | 1/2 | 1 | 2 |
| $k_{\Delta_m} \times \delta_1$ | 0 | 0 | 0 |
| $k_{\Delta_m} \times \delta_2$ | -1/2 | -1 | -2 |
| $k_{\Delta_m} \times \delta_3$ | 1/2 | 1 | 2 |
| $k_{\Delta_m} \times \delta_4$ | 1/4 | 1/2 | 1 |
| Rules for $k_{\Delta_m} < k \leq k_{\Delta_{m+1}}$ | If 2, Then 3 | If 1 or 2, Then 3 | If 1 or 2 or 4, Then 3 |
| Alternative Rules | If Not (1 or 3 or 4), Then 3 | If Not (3 or 4), Then 3 | If Not 3, Then 3 |

Table 9: Example of the vector of lower bound scalars $\underline{k}$ and its relation to a translation $\underline{\delta}$ and resulting rules. Values/indexes with distance $\geq 1$ from the maximum translation ($k\delta_3$) are shown in red. $1 \leq m < n$.

### B.3   GENERALITY OF OUR FRAMEWORK

Though it has been mentioned repeatedly, we will reinstate our objective of proposing a general CE generation framework, and proceed to qualify this goal with an in-depth analysis of the potential scope of our framework. This section delves deeper into the flexibility of GLOBE-CE to model specifics, over a variety of model families (DNNs, XGB, SVMs, results in Appendix D.2), as well as

existing works in the model-agnostic domain of GCE research. We will conclude by considering the expansion of our implementation to current CE methods, and even to subsume the AReS algorithm.

Our motivation lies in demonstrating that our translation scaling techniques possess the requisite flexibility to be overlaid on top of existing criteria or algorithms in the CE space. Ideally, there should also exist specific parameter settings whereby our framework is equivalent to previous works, to guarantee at least equal performance. Analogously, Fast AReS is equivalent to AReS when using $\mathcal{RL}$-Reduction, $r = |V|$, and $s = |V|$. Any such reduction of $r$ or $s$ can be discarded if it hinders the performance of AReS, although we observe empirically that the opposite occurs. Similarly, the fact that our translation generation algorithm is completely general allows our framework to subsume any local method by simply setting the scalar $k = 1$ and the datapoints of interest to just the local input.

**Model Specific – Deep Neural Networks**  DNNs are widely known for their backpropagation properties, which permit gradient calculations for the output layer w.r.t. the input. Several works thus utilise gradient descent to locate counterfactuals for such models. Since the goal of our general CE generation method $G(B, \mathcal{X}, n)$ is to locate $n$ suitable GCE translations, we perform gradient descent on the following objective function, which takes both globality and categorical features into account:

$$\mathcal{L}(\underline{\delta}) = \frac{1}{|\mathcal{X}|} \sum_{\underline{x} \in \mathcal{X}} \left(B_0(\mathbf{round}(\underline{x} + \underline{\delta})) + \lambda \mathbf{cost}(\underline{x}, \mathbf{round}(\underline{x} + \underline{\delta})) + \mathbf{cat}(\underline{x} + \underline{\delta})\right),$$

$$\text{where } \underline{\delta}_{\text{opt}} = \arg\min \mathcal{L}(\underline{\delta}).$$

In this context, $B_0$ is the softmax prediction of the undesired class, or equivalently, when moving beyond a recourse setting, the negative of such for the desired class. The **cost** and **round** terms are as previously described, and the additional **cat** term represents a penalty over categorical features in the resulting counterfactuals (we penalise categorical feature values that do not sum to 1). This naturally conforms to the standard framing of the counterfactual problem, and as previously, once $\delta_{\text{opt}}$ is converged to, it can be scaled to represent the final GCEs. For purely continuous datasets, the objective reduces to the minimisation of $\mathcal{L}(\underline{\delta}) = \frac{1}{|\mathcal{X}|} \sum_{\underline{x} \in \mathcal{X}} \left(B_0(\underline{x} + \underline{\delta}) + \lambda \mathbf{cost}(\underline{\delta})\right)$.

**Model Specific – XGBoost Models**  These also provide an interesting direction for efficient CE generation, namely through readily accessible feature importance scores. In the case of a single decision tree, each node contributes a certain amount to the performance of the tree, and if each such contribution is weighted by the particular number of inputs that it influences, feature importances can be computed for the tree as a whole. Averaging the importances across all trees yields the overall importance per feature. We show a second time that model specific information, in this case feature importance, can be incorporated easily into our framework, simply by weighting the probability of each feature value in our random sampling framework with its feature importance.

**Model Specific – Support Vector Machines**  We study here the linear kernel SVM, a model that offers mathematical expressions for the minimum distances or costs to the decision boundary. Knowledge of such is useful in assessing the performance of our framework; the minimum costs discovered by GLOBE-CE should be as close as possible to the theoretical minima provided.

In our problem landscape, we require minimum costs rather than minimum distances, and we proceed with two assumptions: costs are computed as the $\ell_2$-norm of the individual feature costs; and features are continuous for the sake of this analysis. The first assumption is influenced by the fact that, in the context of linear kernel SVMs, $\ell_1$ costs lead to completely sparse solutions (i.e. solutions where just one feature value changes). Though multiple optimal solutions do exist in this context which are not completely sparse, we wish to test the ability of both AReS and GLOBE-CE in finding a single, unique, and optimal GCE, which the minimum $\ell_2$ cost translation uniquely provides.

We thus aim to minimise the cost $\|C\underline{\delta}\|_2$ of a translation $\underline{\delta}$. The feature costs vector is represented as a diagonal matrix $C$, scaling each feature independently before applying the $\ell_2$-norm. Let the original input, counterfactual and translation be $\underline{x}_0$, $\underline{x}$, and $\underline{\delta} = \underline{x} - \underline{x}_0$, respectively. Given that the decision function of the SVM is $y_i = \underline{w}^T \underline{x}_i + b$, where $y = 0$ at the decision boundary, we derive:

$$y = \underline{w}^T \underline{x} + b = 0 \text{ and } y_0 = \underline{w}^T \underline{x}_0 + b$$

$$\implies -y_0 = \underline{w}^T(\underline{x} - \underline{x}_0) = \underline{w}^T \underline{\delta} = \underline{w}^T C^{-1} C \underline{\delta}$$

Recognising this expression as an inner-product between $\underline{w}^T C^{-1}$ and $C\underline{\delta}$, the Law of Cosines gives

$$-y_0 = \|\underline{w}^T C^{-1}\|_2 \|C\underline{\delta}\|_2 \cos\theta \implies \|C\underline{\delta}\|_2 = \frac{-y_0}{\|\underline{w}^T C^{-1}\|_2 \cos\theta} \, ,$$

which is minimised when $\cos\theta = 1$, such that the angle between $C\underline{\delta}$ and $C^{-1}\underline{w}$ is 0, or alternatively, $C\underline{\delta}$ is parallel to $C^{-1}\underline{w}$. This would imply that, upon normalisation, the two would be equivalent:

$$\frac{C\underline{\delta}}{\|C\underline{\delta}\|_2} = \frac{C^{-1}\underline{w}}{\|C^{-1}\underline{w}\|_2} \implies \underline{\delta} = \frac{-y_0}{\|\underline{w}^T C^{-1}\|_2} \times \frac{C^{-2}\underline{w}}{\|C^{-1}\underline{w}\|_2} = \frac{-y_0 C^{-2}\underline{w}}{\|C^{-1}\underline{w}\|_2^2}$$

Noting that $\underline{w}^T C^{-1} = C^{-1}\underline{w}$, since the cost matrix $C$ is diagonal with $C = C^T$ and $C^{-1} = (C^{-1})^T$, we have thus derived the closed-form expressions for minimum cost $\|C\underline{\delta}\|_2$ and translation $\underline{\delta}$:

$$\|C\underline{\delta}\|_2 = \frac{-y_0}{\|\underline{w}^T C^{-1}\|_2} \quad \text{and} \quad \underline{\delta} = \frac{-y_0 C^{-2}\underline{w}}{\|C^{-1}\underline{w}\|_2^2}$$

This translation on the local input $(\underline{x}_0, y_0)$ in fact applies globally, and we provide details regarding our user study in Appendix D.2 and Section 5.2 of the main text, demonstrating that the GLOBE-CE framework recovers minimum cost recourses very close to the theoretical global optima of the SVM.

**Model Agnostic**  Other black box GCE methods that adopt translation based approaches, such as those in Plumb et al. (2020) and Ley et al. (2022), are easily integrated into our framework. Alternatively, we could view our framework as an extension to those works, filling the previous gaps with regards to minimum costs (by scaling translations) and categorical features (through our interpretation).

In reality, a plethora of issues surround past research on GCE translations. The algorithms in (Plumb et al., 2020; Ley et al., 2022) minimise *distance* between initial inputs (post-translation) and target inputs, resulting in a heavy reliance on the distribution of training data. If data lies too far from the decision boundary, GCEs learnt will not be well optimised for cost. In fact, any sparsely populated areas of the data manifold risk under-representation, despite possible significance w.r.t. the model's decision boundary. Furthermore, when the metric being optimised (the distance between datapoints) is not the metric used to evaluate the resulting explanations, tuning the learning process can become very difficult.

However, such problems can be bypassed with the introduction of our method to scale translations. Typically, the outputs of our random sampling framework are not as useful standalone, but embrace their full utility after the scaling process. Of course, our handling of categorical features would too be a great addition to these methods, which currently neglect to address categorical features.

**The Role of Simplicity in our Framework**  The simplicity of our GCE generation process, as compared to earlier approaches (Rawal & Lakkaraju, 2020; Kanamori et al., 2022), serves to highlight the power of the consequent scaling and selection operations (outlined in this Appendix). It is particularly encouraging that we can achieve superior levels of reliability (high accuracy, low cost) and efficiency compared to the baselines of AReS and Fast AReS, which holds promise for the scalability of GLOBE-CE; as datasets and models increase in complexity, there remains a large scope still for our algorithm to improve and adapt.

**Amending our Framework**  As examples, genetic, evolutionary, or simplex (Nelder-Mead) search algorithms could improve both the reliability and the efficiency of the translation generation process as compared to random sampling. In fact, by framing the global search in an analogous sense to the local problem, as we do in Section 3, we leave the door open for any potential groundbreaking local CE research to be applied immediately in the context of GCEs. Another consideration is that a single translation direction will not always guarantee optimality. In fact, true optimality is only *guaranteed* by a single translation in contexts where the decision boundary spans a hyperplane (e.g. SVMs). We accommodate the use of multiple translations in our approach through a greedy maximum accuracy search at the fixed cost of the search, but the presence of many works in this field outlining methods to achieve diversity should be noted, particularly given that our local framing of the global search directly permits the implementation of diversity techniques such as those in Mothilal et al. (2020).

**Subsuming AReS**  We would further argue that the approach in AReS to produce a relatively large number of GCE explanations is one of the main contributors to its computational complexity, though the form of the two level recourse set generally requires as much. Of course, there is no reason that the apriori (Agrawal & Srikant, 1994) search could not also be used in our framework to generate translations for categorical features by influencing the probability of particular feature values in the same manner as the XGBoost feature importance method that we discussed earlier in this Appendix. On top of this, the ability of our framework to impose any number or combination of subgroup descriptors provides a route to essentially subsume the two level recourse set representation that

AReS deploys, which we portray in Figure 1, with the exception (and advantage) that continuous features do not undergo binning.

**Comparing Subgroups**   The other notable detail in Figures 1 and 1 is the direct comparison of the same GCE between subgroups. Comparing GCEs with different feature values or relative scales can make it difficult to determine what, if any, biases are present in the recourses found. Conversely, by applying the same translation *direction* to both subgroups, the specific differences between recourses are much more easily identified, avoiding the metaphorical 'apples to oranges' comparison. We suggest that translations identified within subgroups are made *fluid*, and their reliabilities evaluated on opposing subgroups. Note that all of the representations that we propose in this text (accuracy-cost profiles, minimum cost histograms, mean translations, and CRCs) support such translation fluidity.

## C   AReS Implementation

We use this Appendix to provide further details regarding the implementation of each stage of the AReS workflow. Our implementation of AReS, without improvements, does in fact differ slightly from that proposed in Rawal & Lakkaraju (2020), and as such we will justify our changes herein. We of course acknowledge that this implementation is far from the most efficient possible, though hope that the patterns and improvements we have identified can aid further development of not only this framework, but others in the global counterfactual explanations space.

Of note is the scalability of AReS, which struggled with Default Credit and HELOC, datasets that contain significantly more points to explain ($|\mathcal{X}_{\text{aff}}|$) than German Credit or COMPAS, and significantly more continuous features. Additionally, the proportion of points with positive predictions (roughly 75% for German Credit and 45% for HELOC on average) influences the ease with which AReS finds recourses. For stringent models (those which scarcely predict positively), it would make sense that the vast majority of frequent itemsets generated by apriori are representative of feature value combinations that exist in the inputs with negative predictions, and we might therefore expect to need to generate an enormous number of triples before we can identify successful recourses.

**Stage 1 Contribution ($\mathcal{RL}$-Reduction)**   We remove items with feature combinations that only occur once (e.g., "Sex = Male, Age < 30" has feature combination "Sex, Age"), yielding $|\mathcal{RL}| = \alpha n$ ($0 \leq \alpha \leq 1$) in $\mathcal{O}(n)$ time. This generates an identical ground set $V$, yet saves $(1 - \alpha^2)n^3$ iterations.

**Stage 1 Contribution (*Then-Generation*, $q$)**   At each iteration of $\mathcal{SD} \times \mathcal{RL}$, we filter the data by the If conditions and redeploy apriori to generate Then conditions. The ground set generated here differs from that in AReS, and we observe significant improvements on continuous features.

**Stage 2 Contribution (*V-Reduction*, $r$, $r'$)**   At large $|V|$, evaluation is costly, yet this is a necessary requirement in finding high-performing triples. Fortunately, we can take advantage of two empirical observations: 1) the generation of a large ground set $V$ is relatively cheap and 2) $acc(V)$ saturates far before the whole set has been evaluated. We evaluate a fixed number of triples and form a new ground set in one of two ways: adding each new triple; or adding only triples that increase the recourse accuracy of the new ground set. We denote these $r$ and $r'$, respectively. For example, $r = $ None and $r' = 1000$ returns 1000 evaluations, and may store fewer than 1000 triples.

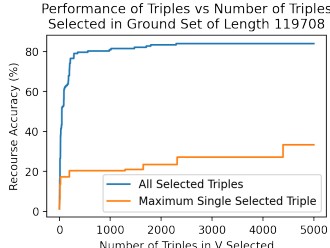

Figure 8: German Credit dataset. Redundancy in triples of ground set $V$.

**Stage 3 Contribution (*V-Selection*, $s$)**   The bottleneck in the AReS framework, however, lies in the submodular maximisation of Stage 3 (Lee et al., 2009). We achieve speedups by further shrinking $V$ pre-optimisation to a more practical starting point. We propose to sort the (new) ground set by recourse accuracy (pre-computed) and select the $s$ highest-performing triples. If $s = r$ or $r'$, no sorting occurs.

### C.1   Ground Set Generation (Stage 1)

AReS includes interpretability constraints for the total number of triples $\epsilon_1$, the maximum width of any Outer-If/Inner-If combination $\epsilon_2$ and the number of unique subgroup descriptors $\epsilon_3$ in $R$. As in AReS, we take $\epsilon_1, \epsilon_2, \epsilon_3 = 20, 7, 10$. Constraints that are independent of the optimisation, such as $\epsilon_2$, are applied in this stage in $\mathcal{O}(n^2)$ and not $\mathcal{O}(n^3)$ time. In our implementation, this involves expediting the $\epsilon_2$ width constraint to the ground set generation process by constraining apriori to

|  | (Stage 1) Ground Set Generation | (Stage 2) Ground Set Evaluation | (Stage 3) Ground Set Optimisation |
|---|---|---|---|
| **AReS** | $n^3$ Iterations Performed | Evaluates Full Ground Set | Searches Full Ground Set |
| **Fast AReS** | $\alpha^2 n^3$ or $n^2 \max_i \mathcal{T}_i$ Iterations Performed | Evaluates and Reduces Full or Partial Ground Set | Searches Reduced and Sorted Ground Set |

Table 10: A summary of our AReS enhancements w.r.t. each stage of the search. We define $\alpha$, $n$ and $\mathcal{T}_i$ below.

only return frequent itemsets that have length $\epsilon_2 - 1$ or less, since those already with width $\epsilon_2$ cannot then be further combined with another itemset to form Outer-If/Inner-If conditions. If the width constraint is not violated for the If conditions, the resulting triple will automatically satisfy the constraint.

The implication of this is that we can apply the constraint in Stage 1, while we generate the ground set (in the first two levels of the iteration through $\mathcal{RL}^3$). This avoids applying the width constraint mid-optimisation in Stage 3, reducing the time complexity of the operation from $\mathcal{O}(n^3)$ to $\mathcal{O}(n^2)$. It also reduces the number of constraints used in Lee et al. (2009), speeding up Stage 3. Since it makes sense that triples which violate the maximum width condition should not be generated in Stage 1, we assume that a similar approach is deployed (though not stated) in Rawal & Lakkaraju (2020).

***Then-Generation*** The apriori algorithm (Agrawal & Srikant, 1994) alluded to in the main text takes a probability threshold $p$ as input. This probability of an itemset in the data, or support threshold $p$, determines the size of $\mathcal{SD}$ and $\mathcal{RL}$, and consequently the size of $V$. Our *Then-Generation* method again utilises application of apriori, requiring a second support threshold $q$. A lower bound for the threshold $q$ is derived here. In fact, there always exists a lower bound when mining frequent itemsets, such as in apriori, since no observed itemset can occur less than once. Thus, setting $q < 1/|\mathcal{X}|$ would be redundant. This allows us to analyse (in Appendix D.1) the effect of $1/|\mathcal{X}| \le q \le 1$. Appendix C.4 further details apriori.

## C.2 GROUND SET EVALUATION (STAGE 2)

The submodular maximisation (Lee et al., 2009) first evaluates the objective function $f$ over all triples $v \in V$, before initialising the solution $R$ as the singleton set $\{v\}$ with the maximum $f(\{v\})$. For large $|V|$, this evaluation becomes computationally costly (more-so does the subsequent ground set optimisation), and many triples are also redundant. However, we require large $|V|$ in order to find high-performing triples and achieve an acceptable upper bound[1] on the final set, $R \subseteq V$.

Our improvement *V-Reduction* evaluates the objective function $f$ (see Appendix C.3) over a fixed number of triples in $V$ (recall that AReS evaluates the entirety of $V$). As we've demonstrated empirically, albeit on the four datasets tried in this investigation, evaluating the entire ground set is wasteful, given that performance of the first $r$ elements of $V$ saturates quickly, and more so if one considers that Stage 3 must then perform submodular maximisation over a space potentially hundreds of times as large, and that (Lee et al., 2009) only guarantees polynomial time.

However, there is a distinction between evaluating the objective function $f$ and evaluating the $acc$ and $cost$ terms used in evaluation. Fortunately, no extra major computation is required to evaluate the $accuracy$ and $cost$ terms, since the objective function $f$ returns model predictions and costs, and although the two processes differ, they can be carried out efficiently in tandem. This is promising, as not only does our method allow us to terminate evaluation once saturation has been reached, but it also provides us with the upper bound $acc(R) \le acc(V)$. In many of our experiments, this upper bound is actually reached in Stage 3 far before the algorithm has completed, presenting us with a straightforward opportunity for early termination of the algorithm. This could further save time dramatically, and provide ease-of-use to practitioners, though was not included in our experiments.

## C.3 GROUND SET OPTIMISATION (STAGE 3)

We introduce two key modifications to Stage 3 of our implementation. The first is to the objective function, the second is to the submodular maximisation in Lee et al. (2009).

---

[1]For instance, if $acc(V) = 25\%$, we cannot achieve $acc(R) > 25\%$; conversely, a ground set with $acc(V) = 80\%$ requires major evaluation and will also include many low-performing, redundant triples.

**Objective Function**   The objective function $f(R)$ in Rawal & Lakkaraju (2020) is designed to be non-normal, non-negative, non-monotone and submodular, and to have constraints that are matroids. These conditions are required for the submodular maximisation in Lee et al. (2009) to have a formal guarantee of convergence. This results in four terms in $f(R)$: *incorrectrecourse*, *cover*, *featurecost*, *featurechange*. Bar the *cover* term, all of these are subtracted from $f(R)$ (i.e., maximising correct recourse by maximising the negative of *incorrectrecourse*). Such an objective function with three adjustable hyperparameters can be very difficult to tune. For that reason, we also trial in our experiments an objective that consists very simply of $acc(R) - \lambda \times cost(R)$, which we maximise. We argue that the formal guarantees of convergence (polynomial time) are largely a misdirection of efforts in the original method. Polynomial time is not particularly helpful when the size of ground sets required for certain datasets/models is huge, and thus we instead focus on reducing the size of the ground set while retaining quality before the submodular maximisation (Lee et al., 2009) is applied.

**Submodular Maximisation**   The algorithm states that, for $k$ constraints, you can exchange up to $k$ elements from your solution set $R$ alongside the addition of one element from $V$. Stated also is that the optimisation should be repeated $k+1$ times, before the best solution for $R$ is then chosen. In reality, both of these induce high computational costs. Trivially, for the latter, ignoring the maximum width constraint (Appendix C.1) and taking $k + 1 = 3$, we will mostly increase the time taken by AReS three-fold. Having observed that both of these steps do not improve the performance of AReS significantly in our experiments, we omit them from the original and improved implementations. Furthermore, since the exchange operation stated is the most costly, our implementation checks for add/delete operations first until such options are exhausted. Note that works such as (Kanamori et al., 2022) instead utilise greedy algorithms, though still require in excess of three hours on simple datasets.

## C.4   APRIORI INTERPRETATION

The apriori algorithm (Agrawal & Srikant, 1994) returns groups of itemsets that are frequently found within a dataset according to some support threshold (the probability of finding such an itemset in the data). Figure 9 demonstrates this (itemsets with less than or equal to 6 features). Rawal & Lakkaraju (2020) states:

1. A recourse set $R$ is made up of triples of the form $(d, c, c')$ denoting an outer if condition ($d$) and inner if-then conditions ($c, c'$), respectively (page 3).

2. "The corresponding features in $c$ and $c'$ should match" (page 3).

3. The optimisation searches for $R \subseteq \mathcal{SD} \times \mathcal{RL}$ (eq. 1, page 5).

4. "If the user does not provide any input, both $\mathcal{SD}$ and $\mathcal{RL}$ are assigned to the same candidate set generated by apriori" (page 5).

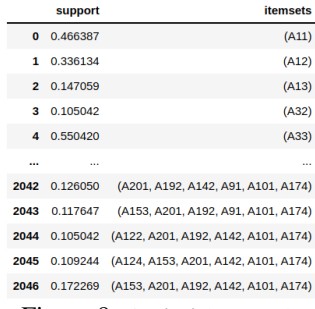

|  | support | itemsets |
|---|---|---|
| 0 | 0.466387 | (A11) |
| 1 | 0.336134 | (A12) |
| 2 | 0.147059 | (A13) |
| 3 | 0.105042 | (A32) |
| 4 | 0.550420 | (A33) |
| ... | ... | ... |
| 2042 | 0.126050 | (A201, A192, A142, A91, A101, A174) |
| 2043 | 0.117647 | (A153, A201, A192, A91, A101, A174) |
| 2044 | 0.105042 | (A122, A201, A192, A142, A101, A174) |
| 2045 | 0.109244 | (A124, A153, A201, A142, A101, A174) |
| 2046 | 0.172269 | (A153, A201, A192, A142, A101, A174) |

Figure 9: Apriori ($p = 0.1$).

We deduce that $\mathcal{SD}$ corresponds to the outer if conditions ($d$), so $\mathcal{RL}$ must then correspond to ($c, c'$). However, from statement 2, apriori cannot provide $c, c'$ because of statement 2, which states that features must match, since a *single* apriori set is incapable of returning the 'Then' part of the recourse rule, yet statements 3 and 4 together imply that apriori generates the full space $R \subseteq \mathcal{SD} \times \mathcal{RL}$.

Furthermore, Lakkaraju et al. (2019) define the search space as $R \subseteq \mathcal{ND} \times \mathcal{DL} \times \mathcal{C}$, where $\mathcal{ND}$ and $\mathcal{DL}$ are analogous with $\mathcal{SD}$ and $\mathcal{RL}$, and $\mathcal{C}$ is the number of classes. The rules then take the intuitive form, with $\mathcal{ND}$, $\mathcal{DL}$ and $\mathcal{C}$ representing each part of the triple. Assuming this reasoning, alongside correspondence with the authors, confirms the form of our search space $R \subseteq \mathcal{SD} \times \mathcal{RL}^2$.

## C.5   FURTHER DISCUSSION

**Customising AReS**   As implicit in (Rawal & Lakkaraju, 2020), our implementation gives the user control over which features are dropped, how continuous features are binned, and the particular subgroup descriptors $\mathcal{SD}$ used for fairness analysis. We additionally posit that the particular data used to generate $\mathcal{RL}$ will potentially affect the final results quite significantly; there should be scope

to assign the 'Then' conditions to an apriori evaluation on the dataset points with positive predictions, as these may be more likely to produce successful recourses. Data scarcity should be taken into account, as small datasets may not contain such a distribution of feature values that allow for effective searches of this nature. Finally, the constraint that all features in the Inner-If and Then conditions must match could possibly be ignored (the CET (Kanamori et al., 2022) and GLOBE-CE frameworks both take this approach).

**Critiquing AReS** We should preface this section by stating that the framework in (Rawal & Lakkaraju, 2020) is an original and major contribution to GCEs, and hope that its limitations can be overcome in consequent research. Save the shortcomings listed in the main text, we find a potential further three for consideration.

Firstly, AReS evaluates its recourses on the same set from which they are learnt (as do we in the main text). In practice, there are many scenarios in which we desire evaluation on a set of unseen test points. We perform such an evaluation, detailed in Appendix D, finding that performance does not deviate significantly, though the GLOBE-CE framework does generalise slightly better.

Secondly, the baselines used to assess AReS are notably weak (e.g. a naive averaging of instance-level explanations), and the effort exerted in tuning such baselines is unknown. While our work considers AReS to be state-of-the-art, owing to its performance against such baselines, making efforts to improve its performance (Fast AReS), perhaps a larger and more varied group of viable baseline frameworks for GCEs that strike a balance between naivety and sophistication could be conceived of.

Finally, the computational expense of the method has not been documented to a strong degree, and the claims made regarding its formal guarantees and framework generality we find not entirely useful. The former claim, while potentially useful, only applies to the optimisation of the ground set $V$ to produce the two level recourse set $R$, providing no concurrent guarantee on the upper bound of $V$, and pertaining also to polynomial time convergence, which scales particularly badly given the extremely high cardinality of $V$ required to achieve acceptable performance. The latter claim, on the generality of the framework, neglects again to account for the size of the optimisation space required by AReS; in throwing a sufficient amount of compute at a simple dataset (far more than the original model required), one should reasonably expect to fit such exhaustive explanations.

**Future Work on AReS** Regarding the first point on generalisation above, we could extend our evaluation on unseen test data to include the effect of overfitting in models and out-of-distribution test points, given the susceptibility of current explanation methods to such inputs.

An interesting property of our Fast AReS optimisation is that there exist a variety of different shrunk ground sets, each with a high performance. While our optimisation simply picks one, multiplicity in AReS might be achieved by shuffling the ground set before the evaluation stage. We suggest the space of possible solutions be explored, tasked with answering the question: *where might this fail?*

Additionally, one might naturally question if a framework such as AReS could be extended beyond recourse, especially to other data forms such as images. While we explain the shortcomings AReS faces on continuous data, we posit that higher level views of image data such as latent spaces or concept embeddings could provide an interesting target for an extended future AReS framework.

Finally, as stated, we bin continuous features into 10 equally sized intervals. This follows the conventions in (Rawal & Lakkaraju, 2020), though we find that this approach struggles to trade performance with efficiency. Another facet of the framework is the direct interpretation of two bins; supposedly, one can move from any point in the first bin, to any point in the second bin, though this is not theoretically confirmed. The use of evenly spaced bins might also be improved upon in certain cases with quantile based discretisation, or more advanced decision tree structures. Lastly, the determination of costs is still a complex and unsolved problem in counterfactual literature– we justify our approach in Appendix E.

## D  EXPERIMENTAL RESULTS

This Appendix further details various experiments on AReS, Fast AReS and our GLOBE-CE framework. User studies, model specific analyses, and hyperparameters are included.

### D.1  AReS OPTIMISATIONS

| | Stage 1 | Stage 2 | Stage 3 |
|---|---|---|---|
| **German Credit** | **OG**: $0.169 \leq p \leq 0.390 \longrightarrow$ 
 **RL**: $0.39 \leq p \leq 0.149 \longrightarrow$ 
 **Then**: $0.9 \leq p \leq 0.303 \longrightarrow$ 
 $q = 0.00125$ | **OG**: $r = 5000$ 
 **RL**: $r = 5000$ 
 **Then**: $r = 5000, q = 0.00125$ 
 **OG**: $0.316 \leq p \leq 0.26, r = |V|$ | **OG**: $0.39 \leq p \leq 0.305, r = |V|$ 
 **RL**: $p = 0.245$ 
 **Then**: $p = 0.48,$ 
 $q = 0.00125$ |
| **HELOC** | **OG**: $0.325 \leq p \leq 0.285 \longrightarrow$ 
 **RL**: $0.325 \leq p \leq 0.203 \longrightarrow$ 
 **Then**: $0.75 \leq p \leq 0.563 \longrightarrow$ 
 $q = 0.000127$ | **OG**: $r = 5000$ 
 **RL**: $r = 5000$ 
 **Then**: $r = 5000, q = 0.000127$ 
 **OG**: $0.325 \leq p \leq 0.3, r = |V|$ | **OG**: $0.324 \leq p \leq 0.318, r = |V|$ 
 **RL**: $p = 0.245$ 
 **Then**: $p = 0.48,$ 
 $q = 0.000127$ |

Table 11: The keys **OG** (*Original AReS*), **RL** ($\mathcal{RL}$-*Reduction*) and **Then** (*Then-Generation)* refer to the generation process of the ground set, as per Section 4. Arrows indicate values carried from one stage to the next. Apriori thresholds $p$ and $q$ are listed. Remaining parameters $r$, $r'$ and $s$ are listed in the Figure 5 plots.

As AReS struggles to achieve sufficient accuracy within reasonable times, we set hyperparameters for *featurecost* and *featurechange*, or $\lambda$, to 0, also finding that the average costs were low and did not vary a large amount, justifying the decision to target correctness. The remaining hyperparameters used in the Figure 5 experiments (Section 5) are as detailed per stage in Table 11. Recall also that we have bounded the range of the apriori threshold $q$ used in *Then-Generation* to $1/|\mathcal{X}| \leq q \leq 1$ (Section 4 and Appendix C.1). Figure 10 demonstrates that for $q > 1/|\mathcal{X}|$, we slightly reduce the runtime, at the expense of a much larger drop in performance. Observe that the red and brown lines (where $p$ is held constant and $q$ is varied) converge to the green and purple lines (where

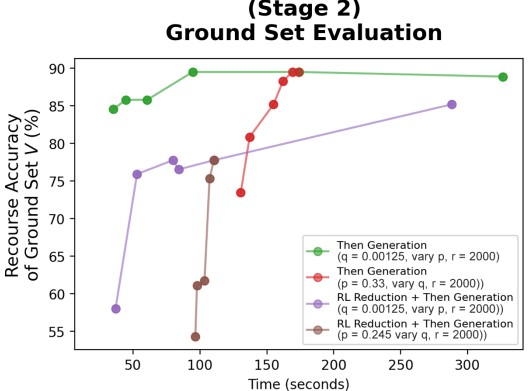

Figure 10: Effect of apriori threshold $q$ in the proposed *Then-Generation* method (German Credit).

$q = 1/|\mathcal{X}|$ and $p$ is varied), respectively. The brown and purple plots also indicate that combining our two improvements $\mathcal{RL}$-*Reduction* and *Then-Generation* performs sub-optimally. We thus decide to evaluate these improvements separately, with a fixed $q = 1/|\mathcal{X}|$ threshold. We note that the choice of $\mathcal{SD} = \mathcal{RL}$ weakens performance, which aids in stress-testing scalability.

### D.2 GLOBE-CE

**Model Specific Analyses** The GLOBE-CE implementations that utilised some sort of model information (DNN gradients, XGB feature importance scores) are depicted in Figures 11 and 12, alongside GLOBE-CE (blue), diverse GLOBE-CE with $n = 3$ (orange) and Fast AReS (green).

The use of **DNN gradient descent** to determine an appropriate translation prior to scaling, shown in red in Figure 11, demonstrates marginally improved performance over GLOBE-CE on continuous data (HELOC, Right), though the same method for categorical data (German Credit, Left) produces

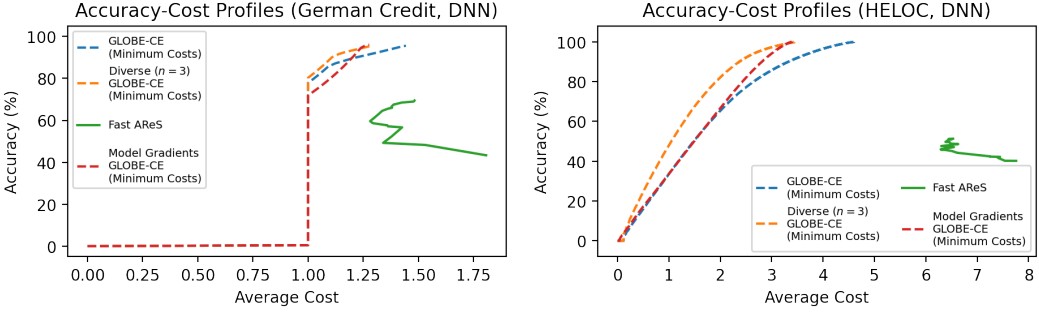

Figure 11: Translations generated by gradient descent and scaled (red), compared to the other frameworks.

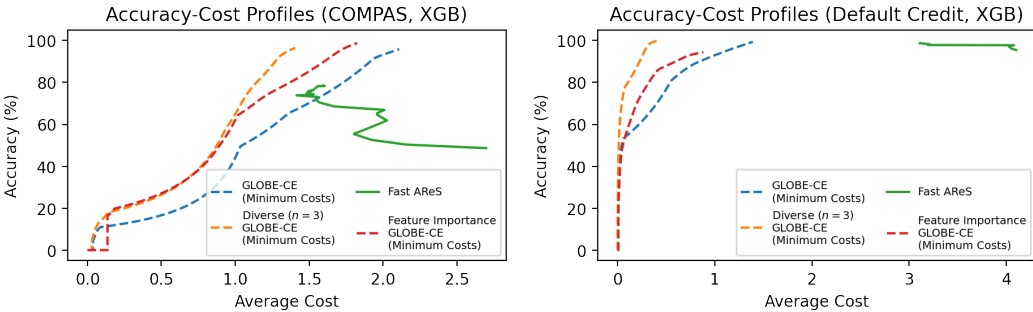

Figure 12: Translations generated by feature importance and scaled (red), compared to the other frameworks.

similar results to GLOBE-CE, with performance dependent on the particular accuracy-cost trade-off desired (as expected, given that gradient descent can struggle in the presence of categorical features).

The use of **XGB feature importances** to weight the probability of the random translation generated prior to scaling, shown in red in Figure 12, demonstrates improved performance over GLOBE-CE on continuous data (Default Credit, Right) and, marginally, on categorical data (COMPAS, Left).

The results across these two particular model classes illustrate the flexibility of our framework and its ability to accommodate model specific properties, doing so effectively with model gradients (DNNs) and feature importance scores (XGB). The effectiveness of the random sampling framework in the absence of any additional model information other than its predictions is also demonstrated.

### D.3    USER STUDIES

In our experiments, participants received either explanations from GLOBE-CE or from AReS, and were subsequently asked to provide bias identification, and a description if they believed a bias to exist. The aim of the study is two-fold. We first hope to establish the importance of providing the underlying distribution of costs as per the GCEs output by a framework (i.e., the proportion of inputs that correspond to each GCE, given that the lowest cost GCE that results in a flipped prediction is selected for each input). Secondly, we hoped to reinforce our claim in Section 2.3 that sub-optimal recourse costs/accuracies can resulting in misleading conclusions regarding bias.

In the first question of the study, both frameworks output the exact same explanations, minus the fact that GLOBE-CE also includes costs and the percentage of inputs for which a particular rule was best. Explanations for AReS/GLOBE-CEs are identical (minus [*square brackets*] for AReS). This is shown below in the Cumulative Rules Chart, where the subgroup *Female* is discriminated against by the model, though due to the data distribution, the subgroup *Male* is discriminated against with respects to the costs required for recourse. This challenges the assumption made in AReS that recourse biases can be simply gauged without knowledge of the inputs affected.

**Cumulative Rules Chart:**

**If** Sex = Male:

> **If** Job = No **and** Property = No,
> **Then** Job = Yes **and** Property = Yes          } Rule M1 [*Cost 2, 90% of Inputs*]

> **If** Healthcare = No,
> **Then** Healthcare = Yes          } Rule M2 [Cost 1, *10% of Inputs*]

**If** Sex = Female:

> **If** Job = No **and** Property = No **and** Savings = No,
> **Then** Job = Yes **and** Property = Yes **and** Savings = Yes          } Rule F1 [*Cost 3, 10% of Inputs*]

> **If** Healthcare = No,
> **Then** Healthcare = Yes          } Rule F2 [*Cost 1, 90% of Inputs*]

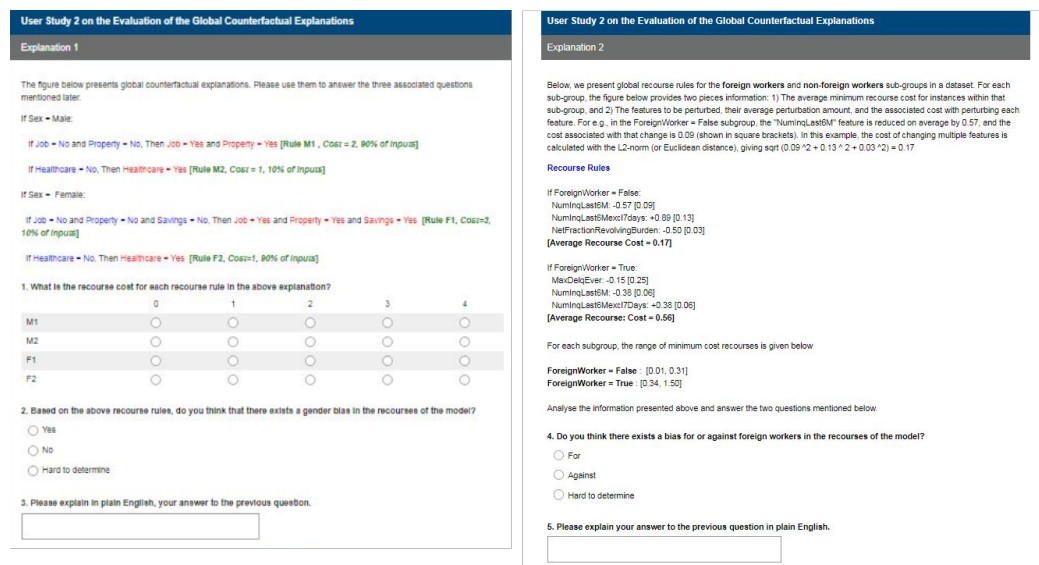

Figure 13: User study 2 snapshots (GLOBE-CE). Left: Explanation 1. Right: Explanation 2. Successful recourse bias identification and description occurs when underlying costs distributions are shown.

In the second question of the study, we introduce a synthetic subgroup *ForeignWorker* to HELOC, and discriminate against it by forcing recourse costs to be higher. We use a linear kernel SVM as our model, in order to compare the recourse costs of AReS and GLOBE-CE against the absolute minimum costs, which we provide a theoretical analysis for in B.3. Our results for GLOBE-CE are as in Figure 1, achieving near-optimal costs, and resulting in correct bias description by users.

# E  DISCUSSION & FUTURE WORK

Having outlined a highly flexible framework for GCEs, it is important that we consider now both the possible limitations of the GLOBE-CE framework, as well as avenues for future work or growth.

## E.1  POSSIBLE GLOBE-CE LIMITATIONS AND FUTURE WORK

**Determining Costs of Actionable Recourse**   We do not unrealistically claim to hold the answers regarding the costs associated with actioning particular counterfactuals. In reality, this is an incredibly complex problem to solve. Not only could the costs of certain actions vary with time, depend on each other and themselves, or be susceptible to unknown degrees of randomness, but they could also depend on a multitude of factors surrounding the specific human being that is ultimately tasked with the action. Disregarding such difficulties, building a model that perfectly reflects a user's struggle in executing a particular set of actions might easily require an improper breach of said user's privacy.

While Rawal & Lakkaraju (2020) proposes the use of the Bradley-Terry model to compute fixed feature costs, and claims that pairwise feature comparisons are "relatively easy for experts to make", neglecting to account for the properties of the end user or even the dependencies between costs will ultimately render such comparisons redundant. Notwithstanding this, searching for *better* cost models remains a worthwhile area of research. For instance, despite its unreliability in this context, the use of the Bradley-Terry model would still surpass in performance the use of non-normalised $\ell_1$ distance as cost. As such, we do not overly focus on cost estimation in this paper, acknowledging that actionability remains, to all intents and purposes, an unsolved problem, and recognising that this could affect the reliability of bias assessment between subgroups based upon costs. Instead, we settle for the use of unit costs between categorical features or per decile of continuous features ($\ell_1$).

**Expanding our Baselines**   The baselines used in AReS are discussed in greater detail in Appendix C.5. Further possible candidates are the GCE translations proposed in (Plumb et al., 2020; Ley et al., 2022), and a non-interpretable accumulation of the *costs* of local CEs, used solely to assess minimum costs per input, and not naively averaged over to produce GCEs. While this last

suggestion doesn't yield GCEs, it could offer a challenging set of minimum costs per input for our framework to seek to outperform.

**Limitations of Categorical Translations**   Our form of categorical translations always yield rules of the form "If A (or B or C ...), Then X" for any one particular feature (with possible negation on the "If" term). However, it may also be useful to represent other forms, such as "If A, Then Not A" or "If A then (B or C)". The minimum costs yielded would not decrease as a result of these new forms, since they can still be represented with the original (e.g. "If A, Then B" would solve simply the first suggestion), though may have interpretability implications which could be explored in future work.

**Alternative GCE Approaches**   The common perspective is to discover CEs from a set of inputs, though a user-based approach which does the opposite could also be proposed, whereby a user specifies a CE of interest, and the group of inputs most strongly affected by such a change are identified and returned. Such an approach could utilise our proposed scaling operation to better summarise the group of inputs in terms of both accuracy and minimum cost, and could also execute without the need for user-specified CEs, generating, scaling and analysing CEs automatically instead.

**Relation between CEs and GCEs**   We postulate the thought experiment that, for a given dataset, there could always conceivably exist further inputs or dimensions at larger or smaller scales whereby global explanations are reduced to local explanations, and vice-versa. For example, a local CE, where all input dimensions are accounted for, can suddenly become a subgroup explanation if extra dimensions are added to the dataset. By the same logic, removing dimensions or adding datapoints can render global explanations local. In the context of fixed models that we find ourselves in here, such modifications possess more theoretical than practical potential, though we pose that such a direction for future research could uncover interested properties between local and global CEs.

**GLOBE-CE: Part II**   We *translate* $N$ negatively predicted instances to $N$ neutrally predicted instances (on the decision boundary). This paves the way for further optimisation of the translation. For example, if the nearest neighbours between the original points and those on the decision boundary do not correspond to the translation for a significant number of inputs, further tuning could ensue.

### E.2   Robustness of Counterfactuals

Recent research has demonstrated instabilities w.r.t. CE techniques from state-of-the-art methods:

- CEs are non-robust (become invalid) in certain scenarios. (Dominguez-Olmedo et al., 2021; Mishra et al., 2021; Dutta et al., 2022)

- Minor perturbations to input features may result in substantially different recourses from popular CE approaches. (Slack et al., 2021)

- Minor changes in the underlying ML model, for example, due to retraining with new data, makes CEs for the old model invalid on the new model. (Upadhyay et al., 2021)

- Noisy human implementation of the suggested recourse may prevent an individual from achieving the desired response from an ML model. (Pawelczyk et al., 2022)

Related research has also identified a positive link between the distance moved past a decision boundary and counterfactual robustness, which suggests that GLOBE-CE can easily be modified to account for robustness by simply increasing the scalar $k$ that is applied to a global translations $\delta$.

