# OpenReview forum: "Global Counterfactual Explanations Are Reliable Or Efficient, But Not Both"
_ICLR.cc/2023/Conference — Submitted to ICLR 2023_

### Official Review · Reviewer_T2G1 · 2022-10-19

**Confidence:** 4
**Correctness:** 3
**Technical Novelty And Significance:** 3
**Empirical Novelty And Significance:** 3
**Recommendation:** 5

**Clarity, Quality, Novelty And Reproducibility:**

Clarity: The paper should be re-structured and reorganized. The overall history and concepts become clear only after several readings.

Quality: The theoretical presentation seems sound. The experimental part lacks statistical significance.

Novelty: the idea of Global CF is limitedly addressed in the literature, but the methods proposed/extended are not disruptive in terms of novelty.

Reproducibility: code is not publicly available, but many details are provided to replicate the experiments.

**Strength And Weaknesses:**

+ A novel counterfactual explanation method is proposed
+ An existing counterfactual explanation method is improved
+ An interesting and limitedly addressed research direction is studied

- The content is not organized in the best possible way, and the narrative is difficult to follow and chaotic in various passages.
- The Fast AReS extension is not clearly explained.
- The experiments are performed on a limited number of datasets and evaluated with a limited and not updated number of measures.


**Summary Of The Paper:**

The paper analyzes various theoretical and technological aspects related to global counterfactual explanation proposing various methods trying to empirically prove the statement "Global Counterfactual Explanations Are Reliable Or Efficient, But Not Both". In particular, it proposes a novel global and efficient counterfactual explainer GLOBE-CE, and improves an existing global counterfactual explainer named Actionable Recourse Summaries (AReS).

**Summary Of The Review:**

The content of the paper might be interesting, but before reaching the qualified level for this conference, the paper should be re-structured and better presented. Experiments should include additional evaluation measures such as those used in Mothilal, R. K., Sharma, A., & Tan, C. (2020, January). Explaining machine learning classifiers through diverse counterfactual explanations. In Proceedings of the 2020 conference on fairness, accountability, and transparency (pp. 607-617) or Guidotti, R. (2022). Counterfactual explanations and how to find them: literature review and benchmarking. Data Mining and Knowledge Discovery, 1-55 or Mazzine R, Martens D (2021) A framework and benchmarking study for counterfactual generating methods on tabular data. Appl Sci 11(16):7274. Also, a comparison against a baseline "dummy" global counterfactual explainers combining in a naive way local counterfactual explainers should be reported.

---

> ### Author Response · Authors · 2022-11-17
> **Response to Reviewer T2G1**
>
> Thank you for the positive feedback. We are glad to see that you found the GLOBE-CE method novel, and that you appreciated the detail provided to allow for reproducibility. We believe that the main contribution of the paper has, however, been misunderstood as a result of the title. To help make the paper clearer and not mislead readers, we have updated the title to *GLOBE-CE: A Translation Based Approach for Global Counterfactual Explanations*.
>
> * **The paper analyzes various theoretical and technological aspects related to global counterfactual explanation proposing various methods trying to empirically prove the statement "Global Counterfactual Explanations Are Reliable Or Efficient, But Not Both".**
>
> While we do demonstrate empirically in the paper that this statement applies to the current state-of-the-art, GLOBE-CE exhibits significant increases in reliability/efficiency. We hope that the title change will clarify that our contribution is in remedying the reliability/efficiency issues that exist in prior works. We will also play down the Fast AReS aspect of the contributions in Section 1 (page 2) i.e. remove its brief mention from the *Improvements* paragraph. While we believe it is strong work, it is meant as an improved baseline and is not the main focus.
>
> * **The content is not organized in the best possible way, and the narrative is difficult to follow and chaotic in various passages.**
>
> As for the narrative, we echo that the title was a possible source for confusion, misleading the reader from the start. The aim is not to empirically prove the quite general reliability/efficiency statement, but to significantly improve on the trade-off shown in the state-of-the-art baselines. With this knowledge, we would kindly ask the reviewer to point to the specific parts of the paper in which the narrative was difficult to follow.
>
> * **The Fast AReS extension is not clearly explained.**
>
> The GLOBE-CE method is the main contribution, and Fast AReS is introduced as an improvement to the baseline of AReS, which itself is too inefficient to allow for a comparison (as demonstrated in the experiments). This was mentioned in the paper:  *"both AReS and our improved version can be used as baselines."* Given this, we believe that the limited space in the paper should be devoted mainly to the GLOBE-CE framework.
>
> * **Code is not publicly available, but many details are provided to replicate the experiments.**
>
> We are happy to add that code will be made publicly available (separate response).
>
> * **The experimental part lacks statistical significance... experiments are performed on a limited number of datasets and evaluated with a limited and not updated number of measures... Experiments should include additional evaluation measures such as those used in Mothilal, R. K., Sharma, A., & Tan, C. (2020, January).**
>
> Thank you for the feedback regarding experiments. Addressing the points one by one:
> * We respectfully disagree that the number of datasets used is limited, having made efforts to select four diverse use cases that would represent the range of data found in real-world applications for recourse. This included testing the methods' abilities on mostly categorical/mostly continuous/mixed type data, as detailed in Table 2.
> * We are unsure that the evaluation of measures such as those in the DiCE paper would be of relevance, as the goal of the paper is to evaluate the reliability/efficiency profile of our method. While we do mention that our framework is flexible to adapt to arbitrary desiderata, the same is not true of the baselines, making a direct comparison hard.
> * On the question of statistical significance: GLOBE-CE achieves speedups orders of magnitudes faster than AReS (up to 200 times on HELOC while simultaneously boosting accuracy by over 80%). We would be grateful to hear what could be done in terms of dataset/model choice to improve on this.
> * AReS does compare its method against "naive" baselines comprised of local CE summaries, as suggested by the reviewer. This performed poorly: averaging local explanations did not perform well. For that reason, we did not duplicate the analysis, focusing on the goal of outperforming AReS.
> * To help persuade the reviewer, we have run additional experiments on the **HELOC** dataset using the DiCE package, generating a local CE for each input (note that this is a bound on the performance of local CEs, not a combination of local CEs). DiCE achieves high accuracy (generating a successful CE for each input), though struggles with cost and computation time. Indeed, one problem with the idea of combining local explainers is that local CEs must be generated in the first place, which is an expensive procedure.
>
> |||DNN|||XGB|||LR||
> |-|-|-|-|-|-|-|-|-|-|
> ||Acc.|Avg. Cost|Time|Acc.|Avg. Cost|Time|Acc.|Avg. Cost|Time|
> |Fast AReS|52%|5.5|109.1s|28%|2.1|93.58s|92%|1.6|127.3s|
> |dGLOBE-CE|95%|3.8|5.46s|80%|2.4|5.6s|100%|0.5|3.85s|
> |DiCE|99.9%|8.59|1242s|100%|6.17|1066s|100%|6.28|1085s|

---

### Official Review · Reviewer_XGq4 · 2022-10-25

**Confidence:** 2
**Clarity, Quality, Novelty And Reproducibility:** See above
**Correctness:** 2
**Technical Novelty And Significance:** 2
**Empirical Novelty And Significance:** Not applicable
**Recommendation:** 1

**Strength And Weaknesses:**

A caveat for the rest of this review: I am not an expert in explanation. As such, much of the jargon was unfamiliar to me. It's possible that an expert in this area would have been able to make more sense of the paper than I did. That being said: this paper is very unclear to a reader who is just a general expert in machine learning. Indeed, I actually still can't figure out what, precisely, this paper is about.

Some illustrative examples of clarity problems:
1. as far as I can tell, "global counterfactual explanation" is not defined

2. it's not clear what data setup the authors have in mind. I guess these are explanations for predictive models. In section 3.1, the authors write "x in X_sub". In context, I guess this notation is meant to be input features. What space do these features live in? Immediately after, the paper uses x + k*\delta, where \delta is a vector. That would seem to imply that x is a vector of real numbers. Does this method then only work for data where the features are vectors of real numbers?

Similarly: what does subgroup mean in this context? What is a "successful" counterfactual?

These confusions all occur in a single sentence. I believe such things could be clarified pretty easily; the issue is that they are not currently.

3. The takeaway messages from Figures 1 and 3 are totally unclear by looking at the figures. The captions do not explain the purpose of these figures.

4. I can't find any result in the paper that appears to correspond to the claim, "Global Counterfactual Explanations are reliable or efficient, but not both"

**Summary Of The Paper:**

I spent an hour struggling to read this and, truthfully, I'm still not sure what it's about.

**Summary Of The Review:**

I was not able to understand this paper well enough to comment on its technical merits. I believe that this is fundamentally an issue of lack of clarity. This paper would need significant improvements to be published.

---

> ### Author Response · Authors · 2022-11-17
> **Response to Reviewer XGq4**
>
> Thank you for reviewing our paper and providing feedback. We will look to address the questions raised:
>
> 1. The way in which we define global counterfactual explanations is written on page 1: "we posit in this work that a GCE should apply to multiple inputs simultaneously, while maximising accuracy across such inputs". They are the same as counterfactual explanations otherwise.
>
> 2. Input space is represented by $x$, and translations thus also live in this space, as is the common use with counterfactual explanations. The conclusion that the method is limited to vectors of real numbers is technically correct, though this does not limit GLOBE-CE's use to continuous features. We mention our theoretical interpretation of a real-valued vector translation on one-hot encoded categorical features in the abstract and section 1, and provide proofs in Section 3.
>
> Subgroups can be defined arbitrarily- we show examples in the paper such as sub-dividing the dataset by gender or foreign-worker status. A successful counterfactual is one that results in a positive prediction (note that in the context of global explanations, a counterfactual may be successful for some inputs whilst unsuccessful for others, hence the accuracy metric).
>
> 3. We agree that the captions for Figures 1 and 3 are short (please see our response to Reviewer NXG7), though we are having a hard time reconciling that the takeaway messages would be totally unclear, given the feedback we have received from other reviewers.
>
> 4. Please see the separate response addressed to all reviewers regarding the title. The empirical results support the claim for the current state-of-the-art (AReS), though our focus is on improving the trade-off dramatically with our proposed GLOBE-CE framework.
>
> We would be glad to provide further details if any of the content in this response is not clear and requires further discussion.

---

### Official Review · Reviewer_NXG7 · 2022-11-02

**Confidence:** 2
**Correctness:** 3
**Technical Novelty And Significance:** 4
**Empirical Novelty And Significance:** 3
**Recommendation:** 8

**Clarity, Quality, Novelty And Reproducibility:**

The paper is generally well-written, adding more information in figures 1,2, and 3 would be helpful.

**Strength And Weaknesses:**

Strengths:

1. Well-motivated idea with clear illustration, especially Figure 2.


Weakness:
1. Clarity on how the mean cost for the translation vectors is estimated in Figure 1 (right-most figure) is needed.


**Summary Of The Paper:**

The paper proposes a framework for global counterfactual explanations while considering the bias that can arise across different subgroups and the need to allow for different translations. The paper also provides an analysis of categorical feature translations. Emprirical results support the theoretical analysis.

**Summary Of The Review:**

I vote for accepting this paper considering the proposed framework for global counterfactual explanations.

---

> ### Author Response · Authors · 2022-11-17
> **Response to Reviewer NXG7**
>
> Thank you for the detailed review. We are glad to hear that you found the paper well written and the GLOBE-CE method novel. We would value your opinion in our proposed title change (see separate response). We believe that the change helps clarify the main contributions, but please feel free to object if you believe otherwise.
>
> * **Clarity on how the mean cost for the translation vectors is estimated in Figure 1 (right-most figure) is needed.**
>
> The mean cost is computed as the norm of the individual costs, as in common in counterfactual literature (the figure states $\ell_2$ in this case). All costs are shown in square brackets, whilst actual feature changes are shown separately. The cost of changing a feature is normalised by the range of that feature, to account for the different scales present in the dataset. This will be compounded in Section 5.2 where we add the definition of cost used in our experiments as:
>
> *For GLOBE-CE, we define the cost of changing a continuous feature to be proportional to the size of the change, such that a change the size of a single bin-width is 1, according to each bin-width computed in AReS. This allows for direct comparison with the AReS method.*
>
> * **Adding more information in figures 1,2, and 3 would be helpful.**
>
> Thank you for the feedback.
> * We have added in Figure 1 that a) the left-most diagram is conceptual, b) accuracy and cost are defined below, and c) that the cost of changing a feature is normalised by the range of that feature, to account for the different scales present in the dataset.
> * We have briefly added in the Figure 2 caption that we demonstrate, in a grid, conceptual situations of bias/no bias vs sub-optimal cost/accuracy.
> * The content remains similar for Figure 3, but we expand on the wording for each conceptual diagram i.e. "Translations are sampled at a fixed cost with generation process $G$" will replace "Fixed cost sampling, $G$", and similar expansions on the wording for the remaining elements.

---

### Official Review · Reviewer_tDod · 2022-11-03

**Confidence:** 3
**Clarity, Quality, Novelty And Reproducibility:** 1. Clarity
**Correctness:** 3
**Technical Novelty And Significance:** 3
**Empirical Novelty And Significance:** 3
**Recommendation:** 6

**Strength And Weaknesses:**

Strengths:
1. This paper attempts to balance efficiency and reliability for global explanation methods of model behavior, which is a valuable perspective in machine learning practice.
2. As the authors claim, it is the first to mathematically address the direct addition of translation vectors to one-hot encodings in the context of Counterfactual Explanations.
3. Overall, the experiments seem well-designed, and the results are promising. Moreover, the improvements of the proposed framework are decent over other methods and consistent in various datasets and under different settings.

Weaknesses:
1. Since the framework is built on many machine-learning models, it would be too hard to reproduce this work because it misses quite a few details, such as how to adapt to various models via different hyper-settings.
2. Categorical Translations have their limitation, such as omitting the rules of ’If A then B or C‘. Unavoidably, based on Categorical Translations, the presented framework may fail in similar cases.

**Summary Of The Paper:**

1. The efficiency of Actionable Recourse Summaries (AReS) is improved, and a method of Global & Efficient Counterfactual Explanations (GLOBE-CE) is designed to address the stability issues flexibly.
2. The authors provide a mathematical analysis of categorical feature translations and use it in their presented method.
3. Experiments on real-world datasets and user studies verify the proposed method's speed, reliability, and interpretability improvements.

**Summary Of The Review:**

My pre-rebuttal rating is '6: marginally above the acceptance threshold'. I really appreciate the problem that this paper addresses and their effort in balancing reliability and efficiency. My concerns are mainly about the limitations caused by Categorical Translations and reproducibility. Please see the detailed concerns in the 'Weaknesses'. I would be happy to read the responses from the authors and willing to increase the score if my concerns are well-addressed.

---

> ### Author Response · Authors · 2022-11-17
> **Response to Reviewer tDod**
>
> Thank you for the detailed review. We are glad to hear that you found the paper well written and the GLOBE-CE method novel. We would value your opinion in our proposed title change (see separate response). We believe that the change helps clarify the main contributions, but please feel free to object if you believe otherwise.
>
> * **Since the framework is built on many machine-learning models, it would be too hard to reproduce this work because it misses quite a few details, such as how to adapt to various models via different hyper-settings.**
>
> We are happy to say that code will be made publicly available, with appropriate documentation and tutorials e.g. one advantage of our method is that it executes orders of magnitudes faster than AReS, allowing for easy exploration of hyper-settings. Unfortunately, for legal reasons, the codebase could not be included in our original submission, but we greatly look forward to releasing it to the community in the following weeks.
>
> * **Categorical Translations have their limitation, such as omitting the rules of 'If A then B or C'. Unavoidably, based on Categorical Translations, the presented framework may fail in similar cases.**
>
> While we agree that the above is a limitation in terms of the form of categorical translations, such explanations are still achievable by simply having two separate global translations (one for 'If A then B', another for 'If A then C'). This is reflected perhaps in our experiments, where we see the dGLOBE-CE method (n=3 explanations) achieve superior performance to a single translation, on the predominantly categorical datasets (COMPAS and German Credit). Note that AReS also does not have the capacity to cover such rules without explicitly increasing the number of explanations generated. Moreover, AReS is actually more restrictive in that it is limited to rules of the form 'If A then B', requiring additional If-conditions to be added as separate rules (our method does allow for 'If A or B, then C').
>
> GLOBE-CE covers both categorical and continuous datasets, witnessing the highest performance (experiments section), whereas the current state-of-the-art AReS has little capacity to cover continuous feature datasets (HELOC experiments). We are encouraged by the fact that we also achieve superior performance on the categorical datasets, and would hope that future work could investigate the idea of categorical translations further, especially once our code implementation is published online and is easily accessible to the community.

---

### Official Review · Reviewer_e9Jt · 2022-11-04

**Confidence:** 2
**Correctness:** 4
**Technical Novelty And Significance:** 4
**Empirical Novelty And Significance:** 2
**Recommendation:** 5

**Clarity, Quality, Novelty And Reproducibility:**

The paper is very clearly written (although again, in my opinion, skims over too many details for me to get a super strong picture of what's happening). I don't know the related literature to assess the degree of novelty, but it appears to be novel. The quality of what I was able to fully understand was strong, but the rigor / reproducibility were difficult to assess when most details were delegated to the appendix.

**Strength And Weaknesses:**

Strengths: The key idea behind GLOBE-CE -- that if groups of instances are explained together by a single set of counterfactuals based on a translation in the input space -- is clever, and, to my knowledge, novel. The experiments seem to show that GLOBE-CE provides a good tradeoff between reliability and efficiency. The paper is very clearly written. The insights about categorical features are also interesting.

Weaknesses:
- There are, in my opinion, too many contributions in this paper for me to fully understand all of them without going through the appendix (and related work) in some detail. Between all the contributions, there appears to have not been room for exposition, definitions, and evidence. For instance, the concept of "recourse cost" is never precisely defined, though identifying bias in it appears to be the phenomenon of study that the methods proposed are meant to uncover. Another example is that the explanation of how AReS works is not really sufficient to follow the paper's contributions to it; I'd have to go read the original paper to understand how it works, and without that intuition, the statement that "analyses suggest that AReS is highly dependent of the ground set V" isn't very convincing.
- The takeaway from the small user study is not obvious to me; measuring the effectiveness of the tool deserves a more in-depth evaluation than is provided here. (However, in a paper introducing this method I would not consider such a user study necessary.)
- I also honestly think the title is somewhat misleading; I thought this paper would quantify reliability and efficiency and either empirically or theoretically examine those tradeoffs. Instead, the paper consists of several contributions vaguely connected by the theme of tackling reliability and efficiency.

**Summary Of The Paper:**

The authors tackle the problem of "global counterfactual explanations," which extend methods from local counterfactual explanations (of individual data instances) to draw global insights about a model. They propose a new framework, which they call GLOBE-CE, which is based on feature translations of vectors of variable length, and provide a mathematical analysis of the implications of this for categorical features. They also improve the efficiency of an existing one (ARES), and compare the two with real world datasets and user studies.

**Summary Of The Review:**

I think there's some strong, interesting, and important work in this paper, but the missing context that seems to have been expected of the reader made it difficult to evaluate. With more pages devoted to the details, I believe this would be a very good paper. But I am not thrilled about its current form.

---

> ### Author Response · Authors · 2022-11-17
> **Response to Reviewer e9Jt**
>
> Thank you for the detailed review. We are glad to hear that you found the paper well written and the GLOBE-CE method novel (input-space translations). We would add that most novelty lies in the idea of varying the length of the translation per input (something prior work did not consider).
>
> * **With more pages devoted to the details, I believe this would be a very good paper.**
>
> Thank you for the positive feedback. We agree that more pages would allow for a complete illustration of events, though we believe that the key contribution of the paper has been misunderstood as a result of the title. To help make the paper clearer and not mislead readers, we have updated the title to *GLOBE-CE: A Translation Based Approach for Global Counterfactual Explanations*.
>
> * **I also honestly think the title is somewhat misleading**
>
> We value and agree with your opinion. Indeed, the title was source for much debate amongst the authors. Having updated it as above, we hope to eliminate the sources of confusion made right from the start. It should be clearer now to readers that GLOBE-CE is the main contribution, and Fast AReS is introduced as an improvement to the baseline AReS, which is simply too inefficient to allow for a baseline comparison (as demonstrated in the experiments).
>
> * **Between all the contributions, there appears to have not been room for exposition, definitions, and evidence. For instance, the concept of "recourse cost" is never precisely defined, though identifying bias in it appears to be the phenomenon of study that the methods proposed are meant to uncover.**
>
> Thank you for pointing this out. We do mention *cost* in the introduction, which is used interchangeably with *recourse cost* in the literature. In terms of the ways in which one can compute this metric, there are many, and we do not limit the method to any particular one. We will add this short definition to Section 5.2:
>
> *For GLOBE-CE, we define the cost of changing a continuous feature to be proportional to the size of the change, such that a change the size of a single bin-width is 1, according to each bin-width computed in AReS. This allows for direct comparison with the AReS method.*
>
> * **The statement that "analyses suggest that AReS is highly dependent of the ground set V" isn't very convincing.**
>
> While this is shown in a) our appendix experiments, b) alternative works that also report 3 hour runtimes for AReS on small datasets, and c) the formal guarantees given in the AReS paper (cubic with respect to the size of the ground set V), we agree that without sufficient space in our text, readers have to either take this at face value or refer to the original paper.
>
> However, since the intention of Fast AReS is to be an improved baseline on the current state-of-the-art (AReS), we would argue that its inclusion is useful but not necessary in comparing our method to AReS. As detailed above, we hope now that the title change will help to clarify the paper's contribution (GLOBE-CE).

---

### Author Response · Authors · 2022-11-17
**Message to Reviewers**

Thank you for the constructive feedback. We have two main updates following the reviews:

* The title has been updated to **GLOBE-CE: A Translation Based Approach for Global Counterfactual Explanations.** Two reviewers understood that GLOBE-CE was the main contribution, and we believe that this would have been clear in the paper for the remaining reviewers, had the title not been misleading from the start.

* We are happy to say that **code will be made publicly available**, with appropriate documentation and tutorials. Due to ongoing compliance checks, the codebase could not be included in our original submission, but we greatly look forward to releasing it to the community in the future.

---

### Decision · Program_Chairs · 2023-01-20

**Decision:**

Reject

**Justification For Why Not Higher Score:**

The presentation issues are significant enough for a major revision & resubmission.

**Justification For Why Not Lower Score:**

N/A

**Metareview: Summary, Strengths And Weaknesses:**

The papers tackle the problem of "global counterfactual explanations," which extend methods from local counterfactual explanations (of individual data instances) to draw global insights about a model. All the reviewers agree that the paper contains some interesting ideas, however, all of them also brought up that the presentation/structure of the paper impairs both the understanding of the contribution as well as the reproducibility and the issue is significant enough for several of the reviewers not to be in support of accepting the paper. I encourage the authors to do a major revision of their paper in light of the reviews and resubmit to another top tier venue.